



# Aviation contrail climate effects in the North Atlantic from 2016-2021

Roger Teoh[1], Ulrich Schumann[2], Edward Gryspeerdt[3], Marc Shapiro[4], Jarlath Molloy[5], George Koudis[5], Christiane Voigt[2,6] and Marc E.J. Stettler[1]

[1]Centre for Transport Studies, Department of Civil and Environmental Engineering, Imperial College London, London, SW7
5  2AZ, United Kingdom
[2]Institute of Atmospheric Physics, Deutsches Zentrum für Luft- und Raumfahrt, 82234 Oberpfaffenhofen, Germany
[3]Grantham Institute – Climate Change and Environment, Imperial College London, London, SW7 2AZ, United Kingdom
[4]Orca Sciences, 4110 Carillon Point, Kirkland, WA 98033, United States
[5]NATS, 4000 Parkway, Whiteley, Fareham, Hampshire, PO15 7FL, United Kingdom
10  [6]Institute of Atmospheric Physics, University Mainz, 55099 Mainz, Germany

*Correspondence to*: Marc E.J. Stettler (m.stettler@imperial.ac.uk)

**Abstract.** Around 5% of anthropogenic radiative forcing (RF) is attributed to aviation $CO_2$ and non-$CO_2$ impacts. This paper quantifies aviation emissions and contrail climate forcing in the North Atlantic, one of the world's busiest air traffic corridors, over 5 years. Between 2016 and 2019, growth in $CO_2$ (+3.13% per annum, p.a.) and nitrogen oxide emissions (+4.5% p.a.) 15  outpaced increases in flight distance (+3.05% p.a.). Over the same period, the annual mean contrail cirrus net RF (204 – 280 mW m$^{-2}$) showed significant interannual variability caused by variations in meteorology. Responses to COVID-19 caused significant reductions in flight distance travelled (-66%), $CO_2$ emissions (-71%), and the contrail net RF (-66%) compared to the prior one-year period. Around 12% of all flights in this region cause 80% of the annual contrail energy forcing, and the factors associated with strongly warming/cooling contrails include seasonal changes in meteorology and radiation, time of day, 20  background cloud fields, and engine-specific non-volatile particulate matter (nvPM) emissions. Strongly warming contrails in this region are generally formed in wintertime, close to the tropopause, between 15:00 and 04:00 UTC, and above low-level clouds. The most strongly cooling contrails occur in the spring, in the upper troposphere, between 06:00 and 15:00 UTC, and without lower-level clouds. Uncertainty in the contrail cirrus net RF (216 – 238 mW m$^{-2}$) arising from meteorology in 2019, is smaller than the interannual variability. The contrail RF estimates are most sensitive to the humidity fields, followed by 25  nvPM emissions and aircraft mass assumptions. This longitudinal evaluation of aviation contrail impacts contributes a quantified understanding of inter-annual variability and informs strategies for contrail mitigation.

## 1 Introduction

Aircraft emissions of $CO_2$, nitrogen oxides ($NO_x$) and soot particles are major contributors of aviation's climate forcing (Lee et al., 2021). Water vapour and soot particles, which consist of a mixture of non-volatile particulate matter (nvPM) and semi- 30  volatile organic and inorganic particles (Petzold et al., 2013; Kärcher and Yu, 2009; Schumann, 1996; Stettler et al., 2011), can also contribute to contrail formation when conditions in the exhaust plume satisfy the Schmidt-Appleman criterion (SAC)



(Schumann, 1996). Contrails that are formed in ice supersaturated regions (ISSR) may persist and spread over several hours and evolve into contrail cirrus (Schumann et al., 2017). The annual mean contrail cirrus cover can be up to 10% of the sky area in high air traffic regions such as Europe and the US east coast (Burkhardt and Kärcher, 2011). Contrail cirrus absorbs and re-

emits outgoing longwave (LW) infrared radiation at all times and only reflects incoming shortwave (SW) solar radiation during the day (Meerkötter et al., 1999), with the net result being a positive (warming) radiative forcing (RF) (Kärcher, 2018; Lee et al., 2021).

An evaluation of global aviation activity in 2018 estimated that the contrail cirrus is the largest contributor to annual mean RF due to aviation, at 111 [33, 189] mW m$^{-2}$ (95% confidence interval), followed by aviation's cumulative $CO_2$ (34 [31, 38] mW

m$^{-2}$) and annual $NO_x$ emissions (8.2 [-4.8, 16] mW m$^{-2}$) (Lee et al., 2021). Although the nominal contrail cirrus net RF is around three times higher than the $CO_2$ RF, its effect on global mean surface temperature change is likely smaller because contrails cirrus is heterogeneously distributed, mainly warms the upper troposphere, dehydrates the atmosphere, and can reduce the occurrence and cloudiness of natural cirrus which partially offsets its warming effects (Meerkötter et al., 1999; Schumann and Mayer, 2017; Ponater et al., 2021). To account for these second-order effects, the climate forcing of contrail cirrus is also

quantified as the effective radiative forcing (ERF) (Ponater et al., 2021; Lee et al., 2021; Myhre et al., 2013). The global annual mean contrail cirrus net ERF (57.4 [17, 98] mW m$^{-2}$) is estimated to be 47% smaller than the contrail cirrus net RF (111 [33, 189] mW m$^{-2}$), but the contrail cirrus net ERF estimate (57.4 [17, 98] mW m$^{-2}$) is still 67% larger than the $CO_2$ ERF (34.3 [28, 40] mW m$^{-2}$) (Lee et al., 2021). Unlike the spatial homogeneity of $CO_2$ forcing, the climate forcing due to contrail cirrus varies spatially and temporally, depending on the local meteorology, surface and cloud albedo, and air traffic density (Meerkötter et

al., 1999; Stuber et al., 2006; Sanz-Morère et al., 2021; Schumann et al., 2012; Chen and Gettelman, 2013; Burkhardt and Kärcher, 2011; Schumann and Heymsfield, 2017). The annual mean contrail cirrus net RF ranges from 70-360 mW m$^{-2}$ over low-albedo regions such as the North Atlantic flight corridor to over 1 W m$^{-2}$ over high traffic regions with higher albedo, such as North America and Europe (Chen and Gettelman, 2013; Schumann et al., 2015; Schumann and Graf, 2013; Burkhardt and Kärcher, 2011; Bock and Burkhardt, 2019).

While atmospheric conditions determine the formation and persistence of a contrail, the aircraft non-volatile soot number emissions index (EI$_n$) modifies the contrail properties (Kärcher, 2018; Schumann, 1996; Voigt et al., 2021; Bräuer et al., 2021b). In the "soot-rich" regime, where EI$_n$ > $10^{13}$ kg$^{-1}$, and in ice supersaturated conditions, the initial contrail ice crystal number is proportional to nvPM EI$_n$ because these particles act as the primary source of condensation nuclei (Schumann, 1996; Kleine et al., 2018). In "soot-poor" conditions, ice crystal numbers are thought to be constrained by a lower limit of around

$10^{13}$ kg$^{-1}$ due to the presence of organic particles and ambient natural aerosols (Kärcher, 2018). Recent cruise measurements have found that the fraction of aircraft nvPM that activates into contrail ice crystals depends on the ambient temperature (Bräuer et al., 2021a) and confirmed that a lower EI$_n$ reduces the ice crystal number and optical depth ($\tau_{contrail}$) of young contrails (Voigt et al., 2021; Bräuer et al., 2021b). Studies that used a contrail lifecycle simulation have shown that a lower nvPM EI$_n$ can reduce contrail lifetime and climate forcing (Burkhardt et al., 2018; Teoh et al., 2020b; Bock and Burkhardt, 2019;

Schumann et al., 2013). Although the nvPM EI$_n$ is recognised as a critical parameter that initialises the ice particle number in



contrail models, previous estimates (Schumann et al., 2021; Teoh et al., 2020b; Caiazzo et al., 2017; Schumann, 2012) were informed by measurements from a limited number of aircraft-engine types that found the $EI_n$ to be in the range of $10^{14}$ to $10^{15}$ $kg^{-1}$ (Moore et al., 2017; Durdina et al., 2017; Wey et al., 2006; Lobo et al., 2015; Boies et al., 2015). However, recent ground nvPM measurements from the International Civil Aviation Organization (ICAO) show that the $EI_n$ can vary by up to five orders

of magnitude for different power outputs and engine types, ranging between $10^{11}$ and $10^{16}$ $kg^{-1}$ (ICAO, 2021). This large variation in nvPM $EI_n$ by engine type would translate to differences in their respective contrail properties (Jeßberger et al., 2013), but these effects have not yet been comprehensively quantified.

In an earlier study focusing on the Japanese airspace (Teoh et al., 2020b), 2% of all flights are responsible for 80% of the contrail energy forcing, the total energy trapped in the atmosphere by contrails. However, these results were derived from six

weeks of air traffic data, and the relatively small sample size did not allow for the identification of factors that led to the formation of strongly warming/cooling contrails. In this study, we use a new dataset containing air traffic data for the North Atlantic region from January 2016 to March 2021 to address these limitations. This dataset also spans the COVID-19 period, which resulted in a 60% year-on-year drop in global passenger traffic (ICAO, 2021), thereby enabling us to quantify the changes in aircraft emissions and contrail climate forcing in this region and compare it with previous estimates (Schumann et

al., 2021; Gettelman et al., 2021; Quaas et al., 2021).

This paper aims to: (i) evaluate the magnitude and the changes in aircraft $CO_2$, $NO_x$ and nvPM number emissions and contrail climate forcing in the North Atlantic from January-2016 to March-2021, including effects from the COVID-19 pandemic ; (ii) identify the set of factors that lead to strongly warming/cooling contrails; (iii) conduct uncertainty and sensitivity analyses relating to meteorological, emissions, and model parameters; and (iv) discuss the implications for contrail mitigation.

## 85  2 Materials and methods

This section describes the datasets and models that are used to simulate the aircraft emissions, contrail properties and climate forcing that arise from individual flights in the North Atlantic region. In summary, we used: (i) an air traffic dataset provided by the UK air navigation service provider, NATS; (ii) the ERA5 high-resolution realisation (HRES, or "reanalysis") and 10-member ensembles reanalysis from the European Centre for Medium-Range Weather Forecast (ECMWF); (iii) the Base of

Aircraft Data (BADA) aircraft performance models from Eurocontrol (2016, 2014); (iv) the ICAO Aircraft Emissions Databank (EDB) (ICAO, 2021) and methodologies to estimate the nvPM (Teoh et al., 2019, 2020b), $NO_x$ (DuBois and Paynter, 2006) and $CO_2$ emissions at cruise; and (v) the contrail cirrus prediction model (CoCiP) (Schumann, 2012).

### 2.1 Air traffic dataset

The air traffic data contains waypoint data for all civil flights that traversed the Shanwick Oceanic Air Traffic Control Area

(OACC) from 1-January-2016 to 31-March-2021, totalling 2.1 million flights. Flight data includes the call sign, operator, origin and destination airports, and the ICAO aircraft type designator. Waypoint data includes 4D positions (longitude, latitude,





altitude, and time) recorded when the flight passes through a series of fixed waypoints along the route and when a climb/descend air traffic command is executed by the pilot in between the fixed waypoints.

There are spatiotemporal heterogeneities in the dataset: waypoints for westbound flights are only recorded in the Shanwick

OACC (10°W to 40°W), while for eastbound flights, additional waypoints are available prior to entry into Shanwick (10°W to 70°W); and the temporal resolution between the recorded waypoints ranges from 30 to 3600 s. We resample the dataset by performing a great circle interpolation between waypoints to produce flight segments with a uniform temporal resolution (60 s). Full flight trajectory coverage in the adjacent Gander OACC is approximated by a great circle interpolation between the first (final) recorded waypoint and the origin (destination) airport, and then removing waypoints outside Shanwick and Gander.

Therefore, the processed dataset approximates the air traffic activity in the North Atlantic flight corridor (50°W, 40°N, 10°W, 75°N). Further information on the NATS dataset is detailed in the Supporting Information (SI) §S1.

**2.2 Meteorology**

Meteorological and radiation data are downloaded from the ECMWF Copernicus Climate Data Store (ECMWF, 2021). The ERA5 HRES reanalysis is obtained at a spatiotemporal resolution of 0.25° × 0.25° × 37 pressure levels × 1 h, while the ERA5

10-member ensembles (0.5° × 0.5° × 37 levels × 3 h) provides an estimate of uncertainties in the reanalysis (Hersbach et al., 2020) and is used to evaluate uncertainty in contrail forcing due to meteorology (Sect. 2.6). For each waypoint, we use a quadrilinear interpolation to estimate the local meteorological values (Schumann, 2012).

The relative humidity with respect to ice (RHi) determines the contrail persistence and lifetime (Schumann, 1996; Kärcher, 2018). However, existing studies have highlighted that the ERA5 humidity fields are generally weakly supersaturated (RHi ≈

100%) and underestimate regions with very high supersaturations (RHi > 120%) (Gierens et al., 2020; Schumann et al., 2021; Rädel and Shine, 2010; Reutter et al., 2020; Tompkins et al., 2007). We present a comparison of the ERA5 humidity fields with in-situ measurements from the In-Service Aircraft for a Global Observing System (IAGOS) database (Petzold et al., 2020; Boulanger et al., 2022) in the SI §S3, and our analysis supports these earlier findings (Fig. S9a). To address these issues, we scale the ERA5 humidity fields throughout the domain by division of coefficient $a$; and then data points within ISSRs (RHi >

100%) are scaled up using a power-law function with coefficient $b$,

$$
\text{RHi}_{\text{Corrected}} = \begin{cases} \dfrac{\text{RHi}}{a} & , \text{when } \left(\dfrac{\text{RHi}}{a}\right) \leq 1 \\[3mm] \min\left(\left(\dfrac{\text{RHi}}{a}\right)^{b}, 1.65\right) & , \text{when } \left(\dfrac{\text{RHi}}{a}\right) > 1 \end{cases}
\tag{1}
$$

The coefficients ($a = 0.9779$ and $b = 1.635$ for the ERA5 HRES; and $a = 0.9666$ and $b = 1.776$ for the ERA5 10-member ensembles) and are found by minimising the Cramer-von Mises test statistic (Parr and Schucany, 1980), a measure of the goodness-of-fit between two empirical distributions, between probability distribution of $\text{RHi}_{\text{Corrected}}$ and that of the IAGOS observations in the spatial domain of the air traffic dataset in 2019, comprising 262 flights and 43919 data points. Further

details are included in the SI §S3. We explore the sensitivity of contrail predictions to meteorological data in Sect. 2.6 and 3.5.





### 2.3 Aircraft performance and fuel consumption

The BADA Family 3 and 4 aircraft performance models are used to simulate the physical forces that act on an aircraft and resultant fuel consumption (Eurocontrol, 2016, 2014). BADA3 covers over 1400 aircraft types, while BADA4, which improves upon BADA3 by including proprietary manufacturer information and accounting for the compressibility and wave drag (Nuic et al., 2010), covers 105 aircraft-engine combinations. The simulated fuel mass flow rate ($\dot{m}_\mathrm{f}$) and net thrust is used to estimate the overall propulsion efficiency ($\eta$), which influences contrail formation (Schumann, 1996).

We use BADA4 where possible, covering 91% of flights in the NATS dataset, and BADA3 for the remaining flights. BADA also provides a range of mass values for each aircraft type. As aircraft mass is not known, we set the initial aircraft mass (at the first waypoint) to the nominal (reference) value for flights occurring before April-2020. For flights affected by the COVID-19 pandemic, from April 2020, we assume a "low" aircraft mass by taking the average of the zero-fuel weight and reference values to account for the reduced load factor. We describe and quantify the sensitivity to these aircraft mass assumptions in Sect. 2.6 and 3.5.

### 2.4 Emissions

An estimate of the nvPM $EI_\mathrm{n}$ and $NO_x$ emissions index ($EI_{NOx}$) requires inputs of $\dot{m}_\mathrm{f}$ and engine specific data. Here, the $\dot{m}_\mathrm{f}$ and aircraft-engine assignments are obtained from BADA, and for available engine types, the ICAO EDB provides the pressure ratio, nvPM $EI_\mathrm{n}$ and $EI_{NOx}$ at the four certification test points (ICAO, 2021).

Three approaches are used to estimate the nvPM $EI_\mathrm{n}$ at cruise: (i) a new approach which utilises nvPM $EI_\mathrm{n}$ measurements from the ICAO EDB to perform a linear interpolation relative to the non-dimensional engine power (66.4% of flights in the NATS dataset); (ii) the fractal aggregates model that was used in earlier studies (Teoh et al., 2020b, a, 2019) to estimate the nvPM $EI_\mathrm{n}$ from the mass emissions index (Stettler et al., 2013; Abrahamson et al., 2016), particle size distribution and morphology (33.3% of flights); and (iii) a constant nvPM $EI_\mathrm{n}$ of $10^{15}$ kg$^{-1}$ when data on the aircraft-engine pair is not available (0.3% of flights). Further methodological details are described in SI §S2. Method (i) is the preferred approach because it captures the distinct emissions profile from different engine types (SI §S2.1). However, the new ICAO EDB nvPM database only provides data for 47 aircraft-engine pairs, thereby necessitating the use of method (ii) or (iii). Method (ii) was formulated based on the emissions profile of singular annular combustors. In a comparison against cruise measurements from Voigt et al. (2021), we found good agreement for the nvPM $EI_\mathrm{n}$ estimated using method (i) for one aircraft-engine pair (SI §S2.3). While limited, this indicates that the emissions indices that are corrected for system line losses are most appropriate.

We estimate the cruise $EI_{NOx}$ using the Fuel Flow Method 2, where $EI_{NOx}$ from the ICAO EDB is fitted linearly on a log-log scale and interpolated using the equivalent $\dot{m}_\mathrm{f}$ at sea level (DuBois and Paynter, 2006). The $CO_2$ emissions are estimated with a constant emissions index of 3.159 kg-$CO_2$ kg$^{-1}$ (Wilkerson et al., 2010).



## 2.5 Contrail simulation

CoCiP is used to evaluate the lifecycle of contrails produced from individual flights (Schumann, 2012). If two consecutive flight waypoints satisfy the SAC, a contrail segment is formed. A Runge-Kutta integration simulates the contrail evolution with time steps of 1800 s until its end of life, defined when the ice number concentration is lower than the background ice nuclei ($< 10^3$ m$^{-3}$), $\tau_{contrail}$ is less than $10^{-6}$, or when the maximum contrail lifetime of 24 h is reached (Schumann, 2012). Further information on CoCiP can be found in the literature (Schumann, 2012; Schumann et al., 2012).

The initial contrail properties are dependent on the nvPM $EI_n$ (Kärcher, 2018; Schumann, 2012). For this study, we calculate the nvPM $EI_n$ for different aircraft-engine types (Sect. 2.4) and set a lower bound of $10^{13}$ kg$^{-1}$ to account for the activation of organic volatile particles and ambient natural aerosols (Kärcher, 2018). We have also used the latest evidence from in-situ measurements of contrails that indicate that the activation of nvPM particles to form contrail ice particles is not complete when ambient temperatures ($T_{amb}$) are near the SAC threshold temperature ($T_{SAC}$) (Bräuer et al., 2021a). Specifically, the proportion of nvPM that activates to form contrail ice crystals is,

$$p_{activation} = -0.661e^{dT_{SAC}} + 1, \qquad \text{if } dT_{SAC} < 0 \qquad (2)$$
$$\text{where } dT_{SAC} = T_{amb} - T_{SAC}.$$

Eq. (2) asymptotically approaches unity and attains values >0.99 for $dT_{SAC} < -4.2$ K.

CoCiP simulates the unique properties and shape of individual contrail segments over time (Schumann, 2012), and accounts for overlapping of contrails above or below clouds present in the meteorological data (Schumann et al., 2012) (Fig. S16). The model is set-up in its original form without humidity exchange between contrails and the background air and radiative effects caused by contrail-contrail overlapping. We acknowledge that the lack of atmospheric interaction and feedback is a limitation of CoCiP relative to general circulation models (Burkhardt et al., 2018; Chen and Gettelman, 2013; Bickel et al., 2019) and is a topic for further research. Previous studies that approximated the contrail-atmosphere humidity exchange in CoCiP have since found a 10-20% reduction in $\tau_{contrail}$ and contrail net RF (Schumann et al., 2015, 2021), while a parametric analysis suggest that the effects of contrail-contrail overlapping to the net RF is small (~0.3 mW m$^{-2}$) in regions such as the North Atlantic and can likely be neglected (Sanz-Morère et al., 2021).

The contrail outputs are saved at a waypoint-, hourly-, flight-, and gridded-level (Schumann, 2012; Teoh et al., 2020b). Contrail and natural cirrus cover in a grid cell are assumed when their respective optical depth ($\tau$) are above a threshold value of 0.1, which corresponds to the satellite detectability threshold (Mannstein et al., 2010). Contrail cirrus coverage (as a percentage of sky area) in a region is defined as the total cirrus cover minus the natural cirrus cover (Schumann, 2012). The local contrail RF (RF', instantaneous change in energy flux per contrail area) (Schumann et al., 2012) of each waypoint is aggregated to obtain the contrail RF in the domain, and the annual contrail ERF is approximated using an ERF/RF ratio of 0.42 (Lee et al., 2021). The contrail energy forcing ($EF_{contrail}$) from individual contrail segments and flights is calculated by,

$$EF_{contrail} = \int_0^T RF'_{net}(t) \times L(t) \times W(t) \, dt, \qquad (3)$$





where $L$, $W$ and $T$ are the contrail length, width, and lifetime respectively. Essentially, the RF', RF and ERF (in W m$^{-2}$) quantify the instantaneous changes in the radiative energy balance at one point in time over a defined spatial domain, whereas EF (in J) provides the cumulative radiative effect of individual contrail segments.

## 2.6 Uncertainty and sensitivity analysis

The simulated contrail properties and climate forcing are sensitive to various input parameters, assumptions, and model
processes. To perform an uncertainty and sensitivity analysis on our results, we re-run the simulation with five distinct setups: (i) the ERA5 10-member ensembles is used to evaluate the uncertainty in contrail properties and forcing due to uncertainty in meteorology and radiation (Hersbach et al., 2020); (ii) a simulation where no corrections are applied to the ERA5 HRES humidity fields; (iii) a simulation to evaluate the sensitivity to $EI_n$, which we assume a constant nvPM $EI_n$ of $10^{15}$ kg$^{-1}$ for all waypoints; (iv) two simulations where we assume a "low" aircraft mass (average of the reference and zero-fuel weight), and
"high" aircraft mass (average of the reference and maximum take-off weight) for all flights at the initial waypoint; and (v) we test the sensitivity to the soot activation near threshold conditions by assuming that all nvPM particles activate to form contrail ice crystals, i.e., $p_{\mathrm{activation}} = 1$. Due to constraints in computational resources, the uncertainty and sensitivity analysis is conducted for 2019 only.

## 3 Results

**3.1 Multi-year statistics**

Table 1 summarises the annual air traffic, emissions, and contrail statistics in the North Atlantic region. Between 2016 and 2019, we found that: (i) the growth in total fuel consumption and $CO_2$ emissions (+3.13% per annum, p.a.) slightly outpaced the total flight distance (+3.05% p.a.) due to the use of larger aircraft types (mean aircraft mass, +1.3% p.a.) (Fig. S6); (ii) total $NO_x$ emissions grew by +4.5% p.a., which is likely attributable to the higher combustor pressure and temperature in more fuel
efficient engines (Kyprianidis and Dahlquist, 2017; Freeman et al., 2018); (iii) the mean nvPM $EI_n$ increased by +0.5% p.a.; and (iv) the contrail cirrus net RF showed significant interannual variability (up to ±19% relative to the mean, ranging between 204 and 280 mW m$^{-2}$), consistent with Wilhelm et al. (2021), indicating a stronger dependence on meteorology than the total flight distance (Fig. S12).





**Table 1: Annual air traffic, emissions, meteorology and contrail statistics in the North Atlantic flight corridor from January-2016 to March-2021.**

| Air traffic and emissions | 2016 | 2017 | 2018 | 2019 | 2020 | Annual growth (2016 – 19) | Pre-COVID: Apr-19 to Mar-20 | COVID: Apr-20 to Mar-21 | % Change: COVID |
|---|---|---|---|---|---|---|---|---|---|
| Total number of flights | 445622 | 462643 | 472303 | 477923 | 203035 | 2.4% | 466336 | 158979 | -66% |
| Total flight distance (x$10^9$ km) | 1.082 | 1.123 | 1.155 | 1.184 | 0.495 | 3.1% | 1.153 | 0.3904 | -66% |
| Total fuel burn (x$10^9$ kg) | 8.13 | 8.41 | 8.65 | 8.92 | 3.45 | 3.1% | 8.69 | 2.48 | -71% |
| Fuel burn per distance (kg km$^{-1}$) | 7.52 | 7.49 | 7.49 | 7.54 | 6.96 | 0.08% | 7.54 | 6.36 | -16% |
| Mean aircraft mass (kg) | 200190 | 200309 | 203050 | 207793 | 193883 | 1.3% | 207788 | 181966 | -12% |
| Mean overall propulsion efficiency, $\eta$ | 0.318 | 0.320 | 0.323 | 0.325 | 0.331 | 0.8% | 0.325 | 0.336 | 3.1% |
| Total $CO_2$ emissions (x$10^9$ kg) | 25.7 | 26.6 | 27.3 | 28.2 | 10.9 | 3.1% | 27.5 | 7.85 | -71% |
| Total $NO_x$ emissions (x$10^9$ kg) | 0.149 | 0.155 | 0.163 | 0.170 | 0.063 | 4.5% | 0.168 | 0.043 | -75% |
| Total nvPM number (x$10^{24}$) | 7.52 | 7.90 | 8.15 | 8.38 | 3.18 | 3.7% | 8.19 | 2.35 | -71% |
| Mean nvPM EI$_n$ (x$10^{15}$ kg$^{-1}$) | 0.92 | 0.940 | 0.943 | 0.939 | 0.922 | 0.5% | 0.943 | 0.947 | 0.4% |

| Contrail properties | 2016 | 2017 | 2018 | 2019 | 2020 | Coefficient of variation (2016-19) | Pre-COVID | COVID | % Change: COVID |
|---|---|---|---|---|---|---|---|---|---|
| Mean ISSR coverage area (%) | 15.8 | 15.5 | 14.9 | 15.2 | 15.2 | 2.4% | 15.3 | 14.9 | -2.3% |
| Flights forming persistent contrails (%) | 55.4 | 56.4 | 52.1 | 54.6 | 47.6 | 3.4% | 50.7 | 50.1 | -1.2% |
| Dist. Forming persistent contrails (%) | 17.3 | 18.4 | 15.3 | 16.2 | 15.9 | 7.9% | 14.7 | 16.3 | 11% |
| Mean contrail age (h) | 3.53 | 3.66 | 3.32 | 3.52 | 3.43 | 3.9% | 3.50 | 3.32 | -5.0% |
| Contrail optical depth, $\tau$ | 0.121 | 0.124 | 0.127 | 0.122 | 0.110 | 2.4% | 0.124 | 0.107 | -14% |
| Contrail cirrus coverage (%) | 0.492 | 0.571 | 0.439 | 0.473 | 0.160 | 11% | 0.420 | 0.141 | -67% |
| Cloud-contrail overlap (%) | 76.0 | 75.1 | 75.1 | 75.0 | 72.4 | 0.65% | 74.5 | 72.2 | -3.2% |
| SW RF (mW m$^{-2}$) | -224 | -258 | -216 | -236 | -69.8 | -7.8% | -206 | -54.6 | -74% |
| LW RF (mW m$^{-2}$) | 449 | 538 | 420 | 471 | 161 | 11% | 410 | 124 | -70% |
| Net RF (mW m$^{-2}$) | 225 | 280 | 204 | 235 | 91.2 | 14% | 204 | 69.6 | -66% |
| Net ERF (mW m$^{-2}$) | 94.5 | 118 | 85.7 | 98.8 | 38.3 | 14% | 85.6 | 29.2 | -66% |
| EF$_{contrail}$ (x$10^{18}$ J) | 60.2 | 74.8 | 54.1 | 62.7 | 24.4 | 14% | 54.3 | 18.6 | -66% |
| EF$_{contrail}$ per flight distance (x$10^8$ J m$^{-1}$) | 0.556 | 0.666 | 0.468 | 0.530 | 0.492 | 15% | 0.471 | 0.475 | 0.9% |
| EF$_{contrail}$ per contrail length (x$10^8$ J m$^{-1}$) | 3.23 | 3.63 | 3.07 | 3.27 | 3.10 | 7.1% | 3.21 | 2.92 | -9.0% |
| Flights causing 80% of total EF$_{contrail}$ (%) | 12.0 | 12.5 | 10.7 | 12.0 | 10.6 | 6.4% | 11.1 | 10.7 | -3.2% |



Comparing statistics during the COVID-19 period (1-April-2020 to 31-March-2021) with the prior one-year period, we find
reductions in total fuel consumption and $CO_2$ emissions (-71%) are greater than the total flight distance (-66%) partly because
we assumed a lower aircraft mass (Sect. 2.3). The proportion of flight distance forming persistent contrails ($p_{contrail}$) during the
COVID period was higher relative to pre-COVID (16.3% vs. 14.7%), partly due to the use of more efficient aircraft types with
a higher mean $\eta$ (+3.1%), which facilitates contrail formation. There is a 66% reduction in the annual contrail cirrus net RF,
but the reduction in SW RF (-74%) is larger than in the LW RF (-70%) for reasons that are consistent with recent COVID
studies (Gettelman et al., 2021; Schumann et al., 2021): reductions in total flight distance and contrail cirrus cover were largest
during the spring months (-80% and -83% relative to pre-COVID, respectively) at a time when the contrail SW RF tends to be
at its maximum (Fig. 1f). We also simulated contrails for the COVID-19 period with pre-COVID traffic to approximate the
likely contrail climate impact under normal traffic conditions and in this scenario, the annual contrail cirrus net RF increased
from 69.6 (actual COVID scenario) to 235 mW m$^{-2}$. To put this in context, if we assume the global contrail climate sensitivity
range of between 0.3 and 0.43 K (W m$^{-2}$)$^{-1}$ (Kärcher, 2018), the surface temperature could be around 0.05 and 0.07 K cooler
than it would been if aviation had not been curtailed due to the pandemic. This approximation suggests that contrail impacts
on surface temperature are likely within the noise levels of the diurnal and seasonal temperature variability in this region,
thereby posing a challenge for measurements/observations to detect these differences (Digby et al., 2021).

Each year, around 12% of all flights in this region account for 80% of the annual EF$_{contrail}$ (Table 1 and Fig. 2), and this is
approximately five times larger than an earlier study which found that 2.2% of flights over Japan caused 80% of the total
EF$_{contrail}$ (Teoh et al., 2020b). Several factors likely have contributed to this difference: (i) the small sample size of the Japan
study with only six weeks of air traffic data; (ii) 58% of all flights over Japan are domestic short-haul flights, meaning that
more time is spent in climb and descent, leading to a smaller mean $p_{contrail}$ (7.2%) relative to the North Atlantic (16.6%, Table
1); and (iii) the use of radar surveillance in the Japanese airspace with more randomised traffic patterns, relative to the organised
track structure (OTS) in the North Atlantic where flight paths often traverse the same air parcel and have a smaller variation
in flight distances and headings.

## 3.2 Seasonal contrail statistics

The air traffic activity, meteorology and contrail characteristics in the North Atlantic exhibit seasonal patterns (Fig. 1). Air
traffic activity peaks in the summer (Fig. 1a), but there is a higher occurrence of persistent contrails during the winter despite
traffic levels being 30% below peak values: (i) $p_{contrail}$ is higher in winter relative to summertime (22.7% vs. 12.5%) (Fig. 1b),
consistent with the $p_{contrail}$ derived from in-situ measurements (Gierens et al., 1999); and (ii) these contrails persist for longer
(3.2 vs. 2.6 h) (Fig. 1c). These phenomena can be attributed to seasonal variations in the ISSR coverage (Fig. S13). In winter,
larger horizontal extent of ISSRs increases $p_{contrail}$; while thicker vertical extent of ISSRs reduces the probability of contrails
encountering warm/dry air after forming and sedimentation, thereby increasing their lifetime (Agarwal et al., 2022; Kärcher,
2018; Schumann and Heymsfield, 2017). The mean contrail ice crystal radius ($r_{ice}$) and $\tau_{contrail}$ in wintertime are around 25%





smaller relative to the summer (Fig. S14g and S14h), which is likely caused by seasonal variations in the tropopause and temperature (Hoinka et al., 1993; Lewellen, 2014): a higher proportion of flights cruise above or close to the tropopause in wintertime because of the lower tropopause height (Fig. S13) and contrails are formed at lower temperatures with less condensable water (Fig. S14d and S14e); while a higher tropopause height in the summer and the moist upper troposphere facilitate the formation of contrails with larger ice water content, $r_{ice}$ and $\tau_{contrail}$.




**Figure 1: Monthly statistics in the North Atlantic region from January-2016 to March-2021, including the (a) total flight distance; (b) percentage of flight distance forming persistent contrails ($p_{contrail}$); (c) mean contrail age; (d) percentage of contrail area overlapping with natural cirrus; (e) contrail cirrus cover; (f) contrail cirrus SW RF; (g) LW RF; and (h) net RF. Additional variables that are not presented in this figure are available in Fig. S14 (SI §S4).**




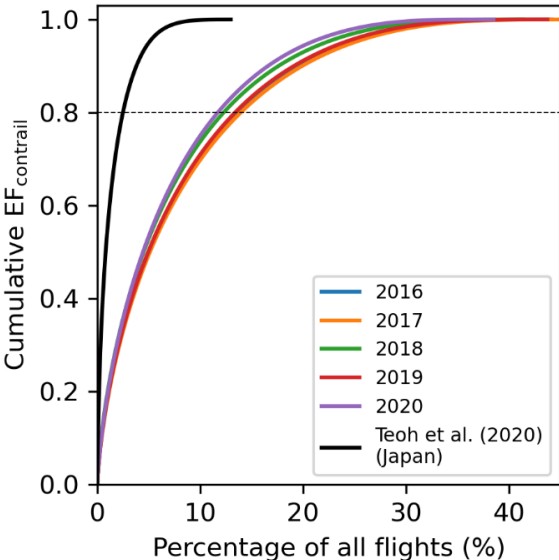

**Figure 2: Cumulative density function of the total EF$_{contrail}$ versus the percentage of flights that accounts for the proportion of EF$_{contrail}$. Individual coloured lines represent the results for each year (2016 – 2020) in the North Atlantic, while the black line shows the results from an earlier study in the Japanese airspace (Teoh et al., 2020b).**

An earlier study highlighted that contrails formed over the North Atlantic can exhibit a net cooling effect under cloud-free conditions (Sanz-Morère et al., 2021). However, our results show that these conditions occur relatively infrequently. The mean ERA5 natural cirrus coverage in this region varies from 40% (summer) to 59% (winter), and up to 90% of the contrail area overlaps with natural cirrus in winter (Fig. 1d). The contrail cirrus coverage generally peaks at around 0.7% in the summer (Fig. 1e), coinciding with the minimum natural cirrus cover and cloud-contrail overlap during this period.

Between 2016 and 2019, the mean contrail cirrus SW RF in springtime is around 16% larger than summertime (-323 vs. -280 mW m$^{-2}$, Fig. 1f) despite air traffic levels being 11% below the summer peak and both seasons having a comparable mean solar direct radiation (SDR) (~394 W m$^{-2}$, Fig. S14i). This can be attributed to summertime meteorological conditions that are generally less favourable for contrail formation ($p_{contrail}$: 13% vs. 17% in spring) and persistence (mean age: 2.65 vs. 3.03 h).

The contrail LW RF is dependent on the surface temperature and outgoing longwave radiation (OLR), contributing to smaller seasonal fluctuations in the LW RF relative to the SW RF (Fig. 1g). When taken together, the mean contrail net RF peaks during the winter (302 mW m$^{-2}$) and is at a minimum in spring (196 mW m$^{-2}$) (Fig. 1h). Figure 1 also shows outliers in $p_{contrail}$, contrail cirrus cover and RF values in April-2017, and this was primarily caused by large intersections between the ISSR and OTS (Fig. S15).





## 3.3 Conditions for forming strongly warming/cooling contrails

Figure 3a shows the contrail net RF and EF$_{contrail}$ per contrail distance for each hour of 2019. There is a clear diurnal effect, where the mean net RF is ~199 mW m$^{-2}$ during daylight hours, increasing to ~385 mW m$^{-2}$ (and up to 2.6 W m$^{-2}$) around ±1 h of the sunrise and sunset time, and then falls to 293 mW m$^{-2}$ at night-time (periods where SDR = 0). The peak net RF around sunrise and sunset is partly attributable to the flight scheduling in this region where eastbound and westbound traffic activity is highest at around 03:00-06:00 and 12:00-16:00 UTC respectively (Fig. S5a). At night, the mean net RF is 24% smaller than the peak values around sunrise and sunset because air traffic is at a minimum between 18:00-02:00 UTC, but contrails that persist during these times have the largest EF$_{contrail}$ per contrail length, which is up to one order of magnitude larger than day-time values (Fig. 3b).

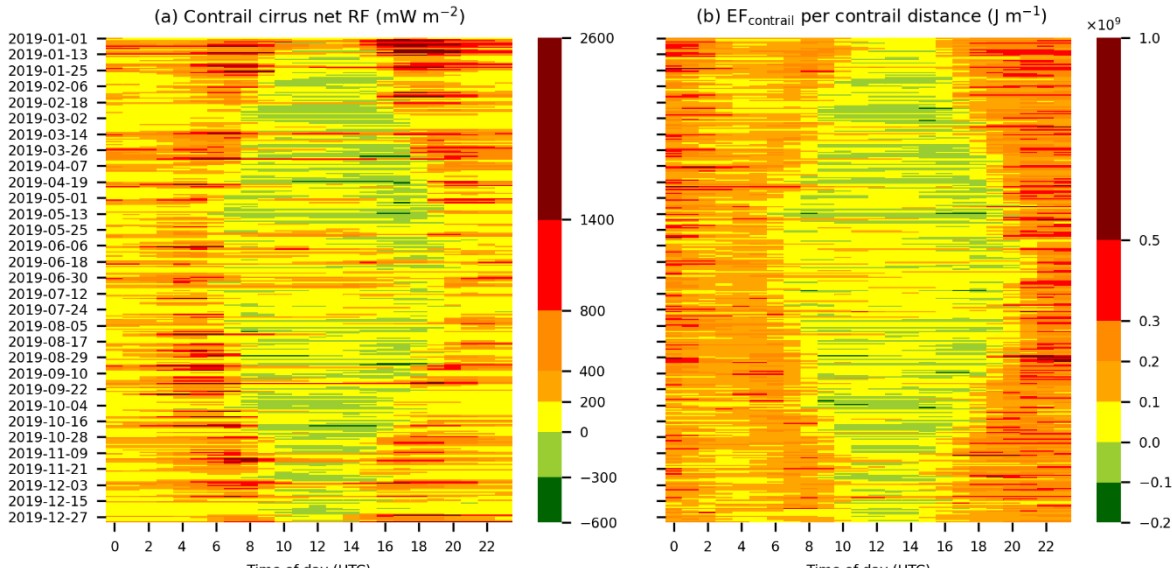

**Figure 3: The regional (a) contrail cirrus net RF; and (b) EF$_{contrail}$ per contrail distance in the North Atlantic flight corridor for each hour of 2019.**

The daytime contrail cirrus net RF also exhibits a large variability (Fig. 3a) and can be strongly warming (1086 mW m$^{-2}$, 18-Sept-2019 14:00) or cooling (-594 mW m$^{-2}$, 28-Aug-2019 10:00). Taking these two one-hour periods as a demonstrative example, they are similar in that the mean RHi and OLR at the contrail waypoints agree within 1.7% and 6.7% respectively (Fig. S17a and S17b). The SDR at 10:00 on 28-Aug-2019 (net-cooling) was 31% smaller than at 14:00 on 18-Sept-2019 (net-warming) (Fig. S17c), which would work in favour of greater (negative) SW RF for the warming period. However, lower percentage of cloud-contrail overlap (68% vs. 76%, Fig. S16) and thus mean albedo along contrail waypoints (0.27 vs. 0.48, Fig. S17d) on 28-Aug-2019, both of which are likely caused by a lower occurrence of low-level optically thick water clouds, led to the different contrail net RF in both periods. A larger cloud-contrail overlap and albedo reduces the SW RF attributable





295  to contrail cirrus (Schumann et al., 2012; Meerkötter et al., 1999; Sanz-Morère et al., 2021), and means that the LW RF dominated at 14:00 on 18-Sept-2019.

The full dataset contains 1.27 million contrail-forming flights, and contrails formed by 79.7% of these flights are warming ($EF_{contrail} > 0$ J). Figure 4 compares the distribution of aircraft, traffic, meteorology, and radiation variables for all contrail-forming flights, and separately for flights with strongly warming ($EF_{contrail} > 99^{th}$ percentile for each year) and cooling ($EF_{contrail}$

300  $< 1^{st}$ percentile) contrails. Here, we show that the differences in $EF_{contrail}$ can be grouped into four categories: (i) seasonal changes in meteorology and radiation; (ii) time of day; (iii) background cloud fields; and (iv) the nvPM number emissions from different aircraft types.

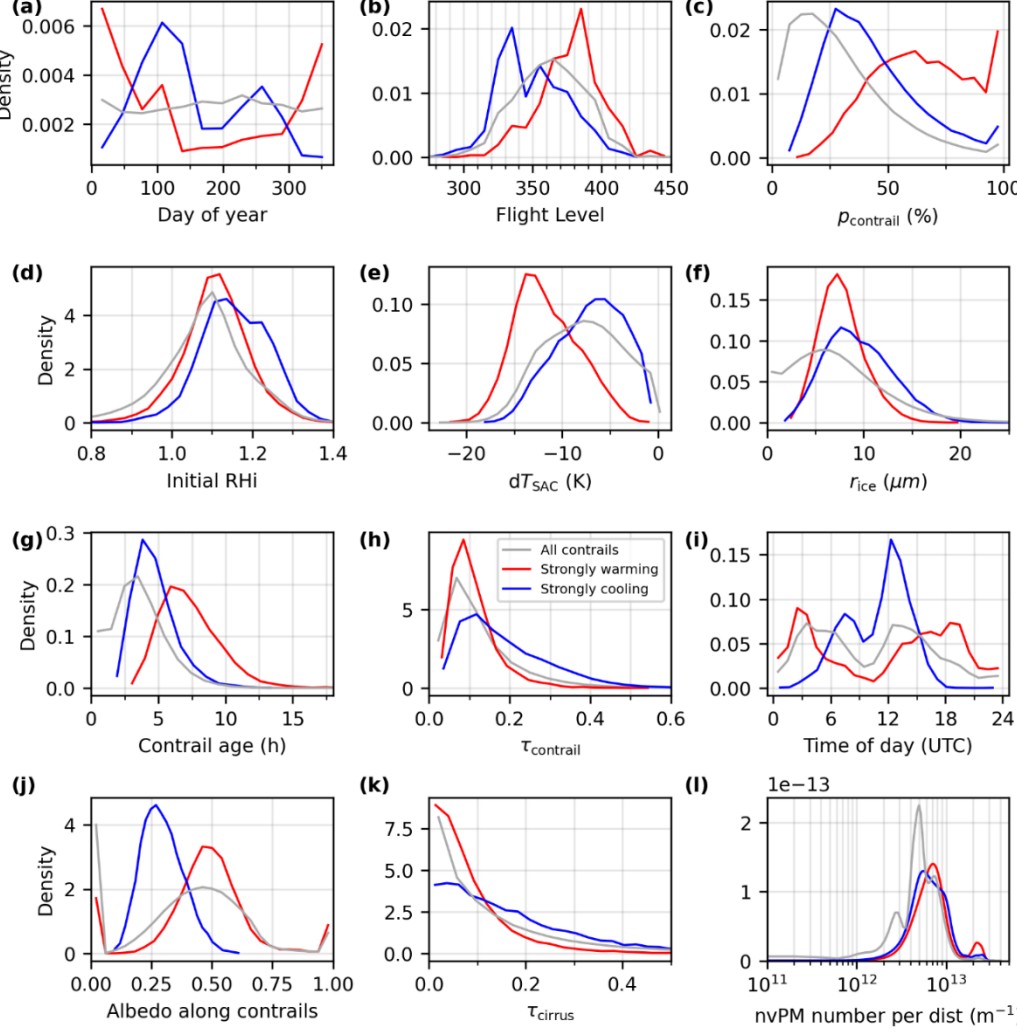

**Figure 4: Probability density function of the aircraft, meteorology and contrail properties for all contrail-forming flights (grey lines),**
305  **flights with strongly warming (red lines, $EF_{contrail} > 99^{th}$ percentile) and cooling contrails (blue lines, $EF_{contrail} < 1^{st}$ percentile). The time-of-day variable in panel (i) represents the time when the flight is at its midpoint between the first and final recorded waypoints.**



### 3.3.1 Seasonal changes in meteorology and radiation

Figures 4a to 4c show that strongly warming contrails are generally formed in wintertime and close to the tropopause (above FL350/35,000 feet) with a mean $p_{contrail}$ of 65%; while cooling contrails are primarily formed in spring and in the upper troposphere (below FL350/35,000 feet) with a mean $p_{contrail}$ (43%) that is larger than all contrail-forming flights (30%). These differences can be attributed to seasonal changes in the ISSR coverage area and tropopause height (Fig. S13a). Conditions close to the tropopause tend to be drier than the upper troposphere, leading to strongly warming contrails having: (i) a smaller amount of condensable water (initial RHi = 111% and $dT_{SAC}$ = -11.7 K, shown in Fig. 4d and 4e) relative to cooling contrails (RHi = 116% and $dT_{SAC}$ = -7.1 K); and (ii) a smaller mean $r_{ice}$ (7.71 vs. 9.47 µm for cooling contrails, Fig. 4f). For all contrail-forming flights, $r_{ice}$ is correlated with the initial RHi (R=0.661), $dT_{SAC}$ (R=0.551), and $\tau_{contrail}$ (R=0.585), and negatively correlated with the contrail age (R = -0.350) (Fig. S18) because $r_{ice}$ influences the ice crystal sedimentation rate. Therefore, the smaller $r_{ice}$ for strongly warming contrails contributes to a larger mean contrail lifetime (7.3 vs. 4.7 h, Fig. 4g) and a smaller mean $\tau_{contrail}$ (0.109 vs. 0.185, Fig. 4h) relative to strongly cooling contrails. A smaller $\tau_{contrail}$ reduces the contrail SW RF' more strongly than the LW RF' in clear sky conditions and for the same surface albedo[15].

Seasonal changes in SDR leads to a high proportion of strongly warming contrails forming in wintertime (Fig. 4a) when SDR is at a minimum (Fig. S14i). In contrast, strongly cooling contrails predominantly occur in the spring rather than in the summer despite both seasons having a comparable mean SDR (~394 W m$^{-2}$, Fig. S14i). This can be attributed to a lower mean OLR in spring (221 vs. 237 W m$^{-2}$ in the summer, Fig. S14j), which reduces the mean contrail LW RF' by 23% (4.9 vs. 6.3 W m$^{-2}$ in the summer), and a larger mean contrail lifetime (3.0 vs. 2.7 h in the summer, Fig. 1c) that can increase the absolute magnitude of $EF_{contrail}$.

### 3.3.2 Time of day

The time of day is a key determinant of $EF_{contrail}$ (Fig. 4i). Flights with strongly warming contrails tend to occur between 15:00 and 04:00 UTC and the mean contrail lifetime of 7.3 h (Fig. 4g) suggests that these contrails spread and persists through the night. In contrast, strongly cooling contrails are predominantly formed by flights that traverse the airspace between 06:00 and 15:00 UTC with a shorter contrail lifetime (4.7 h), thereby maximising the SW RF' and sublimating before nightfall.

### 3.3.3 Background cloud field

Contrails forming above low-level cloud (indicated by a high mean albedo of 0.47, Fig. 4j) are more likely to be strongly warming because the incoming SDR would have been reflected by the low-level cloud regardless of the presence of contrails, thereby reducing the contrail SW RF'. In contrast, strongly cooling contrails are more common over regions with little low-level cloud, giving them a maximum SW RF' through a strong albedo contrast with the dark ocean surface (mean underlying albedo of 0.29, Fig. 4j) and below optically thick high-level cirrus. For these strongly cooling contrails, the mean overlying natural cirrus optical depth ($\tau_{cirrus}$) is around two times larger than for strongly warming contrails (0.172 vs. 0.081, Fig. 4k) and




because the overlying cirrus already has a high LW RF, this reduced the additional LW RF' that can be attributed to contrails, thus suppressing their warming effect.

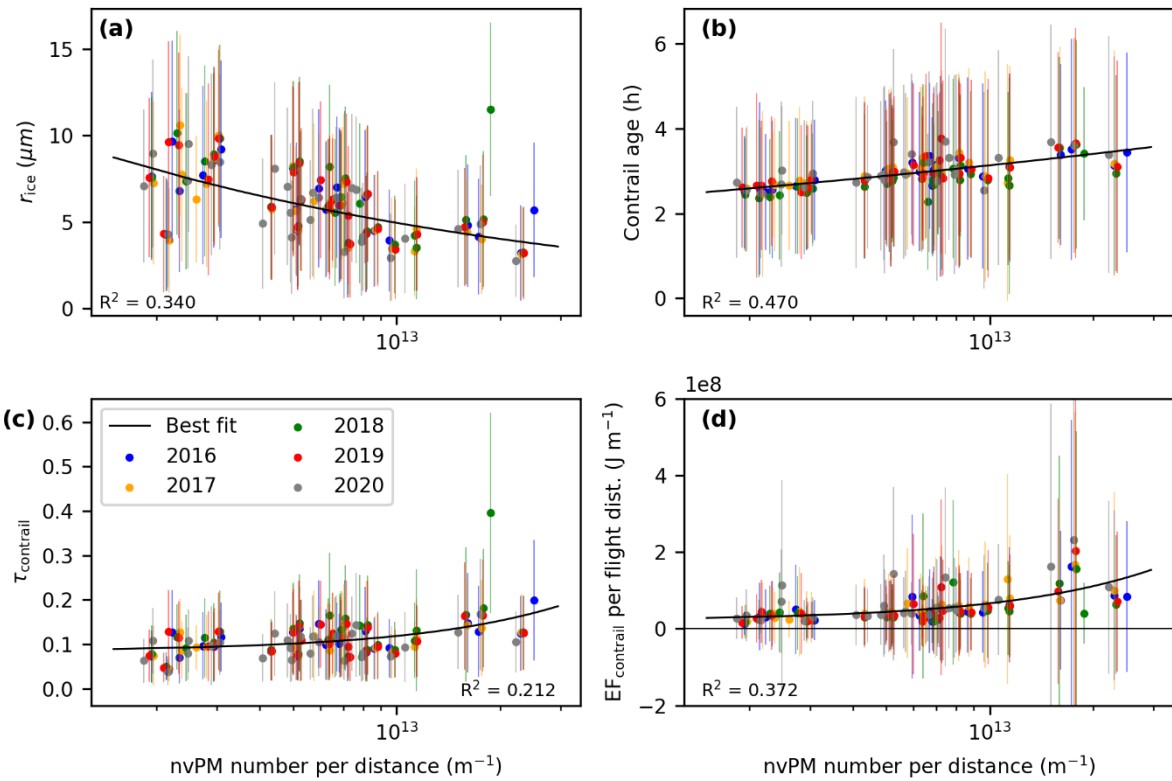

**Figure 5: The (a) contrail ice crystal radius ($r_{ice}$); (b) age; (c) optical depth ($\tau_{contrail}$); and (d) $EF_{contrail}$ per flight distance vs. nvPM number emissions per distance travelled. Each data point represents the annual mean values for different aircraft types, and the error bars represent one standard deviation.**

### 3.3.4 Influence on nvPM emissions

Strongly warming contrails tend to be associated with a higher mean nvPM number emissions per unit distance travelled (1.1 $\times 10^{13}$ m$^{-1}$) that is 14% and 34% larger relative to strongly cooling contrails (8.9 $\times 10^{12}$ m$^{-1}$) and all contrail-forming flights (6.8 $\times 10^{12}$ m$^{-1}$) respectively (Fig. 4l). Comparing the effects of different aircraft types shows that 43.4% (17.4%) of flights with strongly warming (cooling) contrails are powered by one engine-combustor type, the Phase 5 Rich-Quench-Lean combustor, which has one of the highest nvPM $EI_n$ in the ICAO EDB ranging from 0.7 to 1.4 $\times 10^{15}$ kg$^{-1}$ (ICAO, 2021). In particular, while

one specific very large wide-body aircraft is only used for 2.4% of all flights, it accounted for 18.0% (6.4%) of flights with strongly warming (cooling) contrails (Fig. S19 and Table S5). This aircraft has the highest nvPM number emissions per distance (~2.4 $\times 10^{13}$ m$^{-1}$), the product of the nvPM $EI_n$ and fuel consumption, relative to other aircraft types (mean of 3.9 $\times 10^{12}$ m$^{-1}$). This leads to smaller $r_{ice}$ as the fixed ambient vapour is distributed to more particles, longer contrail lifetimes due to lower sedimentation rate of particles with smaller $r_{ice}$, and larger $\tau_{contrail}$ due to the Twomey effect (Fig. 5). Indeed, in-situ contrail



measurements have shown that the $\tau_{contrail}$ resulting from different aircraft types with different nvPM emissions can vary by up to a factor of 4 (Jeßberger et al., 2013). These processes increase the magnitude and variability of $EF_{contrail}$ for aircraft with higher nvPM emissions (Fig. 5d), and the sign of $EF_{contrail}$ depends on a trade-off between $\tau_{contrail}$ and contrail lifetime: a higher $\tau_{contrail}$ increases the SW RF' more strongly than the LW RF' and can lead to a larger cooling effect during the day (Schumann et al., 2012); but a longer lifetime can also cause the contrail to be strongly warming as it spreads and persists into the night.

However, a comparison of $EF_{contrail}$ per passenger-km shows that one specific medium wide-body aircraft has the highest value ($6.7 \times 10^5$ J m$^{-1}$), and that the very large wide-body aircraft mentioned above ($4.5 \times 10^5$ J m$^{-1}$) is close to the median value for the 18 aircraft types considered in Table S5.

### 3.4 Meteorological uncertainties

  The ERA5 ensemble spread represents the observation uncertainties (provided to the data assimilation system) and model state
uncertainties in the reanalysis (Hersbach et al., 2020). We propagate these uncertainties to estimates of the annual emissions and contrail properties for 2019. Figure 6 shows that uncertainties derived from the ensembles are small for the total fuel consumption, $CO_2$, $NO_x$ and nvPM number emissions (within ±0.01% relative to the ensemble mean), while the fleet-aggregated contrail properties have larger uncertainty bounds of up to ±8%. Uncertainty in the 2019 contrail cirrus net RF (216 – 238 mW m$^{-2}$, Table S6) is smaller than the interannual variability between 2016 and 2019 (204 – 280 mW m$^{-2}$, Table 1).

The lower spatiotemporal resolution of the ensembles relative to the HRES leads to differences in the simulated contrail properties. Figure 6 shows that the ensemble mean $\tau_{contrail}$ is 2.6% larger than the nominal HRES simulation, which influenced the SW RF (+31%) more strongly(Schumann et al., 2012) than the LW RF (+12%). Although the range of net RF from the ensembles (216 – 238 mW m$^{-2}$) encompasses the nominal HRES value (235 mW m$^{-2}$), the net RF from nine of ten ensemble members is below the HRES value.

To evaluate the consistency in contrail prediction for specific flights, we compared the set of flights with $EF_{contrail}$ > 95$^{th}$ percentile in the nominal HRES simulation with each ensemble simulation (5% of all contrail-forming flights) and found that 36.9% of these flights have an $EF_{contrail}$ > 95$^{th}$ percentile in all ten ensemble members. The characteristics of flights with strongly warming/cooling contrails in each ensemble member (Fig. S20) are generally consistent with the HRES (Fig. 4). However, unlike the HRES, the ensembles do not predict the occurrence of strongly warming or cooling contrails before dawn
or dusk, respectively (Fig. 4i vs. Fig. S20i). This is likely due to the lower temporal resolution of the ensembles (3 h vs. 1 h for the HRES), potentially causing an overprediction of the mean contrail age (+7.7% relative to the HRES, Fig. 6d) and change in the sign of $EF_{contrail}$ as contrails persist through dawn/dusk. This also caused percentage of flights accounting for 80% of the annual $EF_{contrail}$ in the ensembles (~8.6%, Table S6) to be lower than the HRES (12.0%, Table 1).

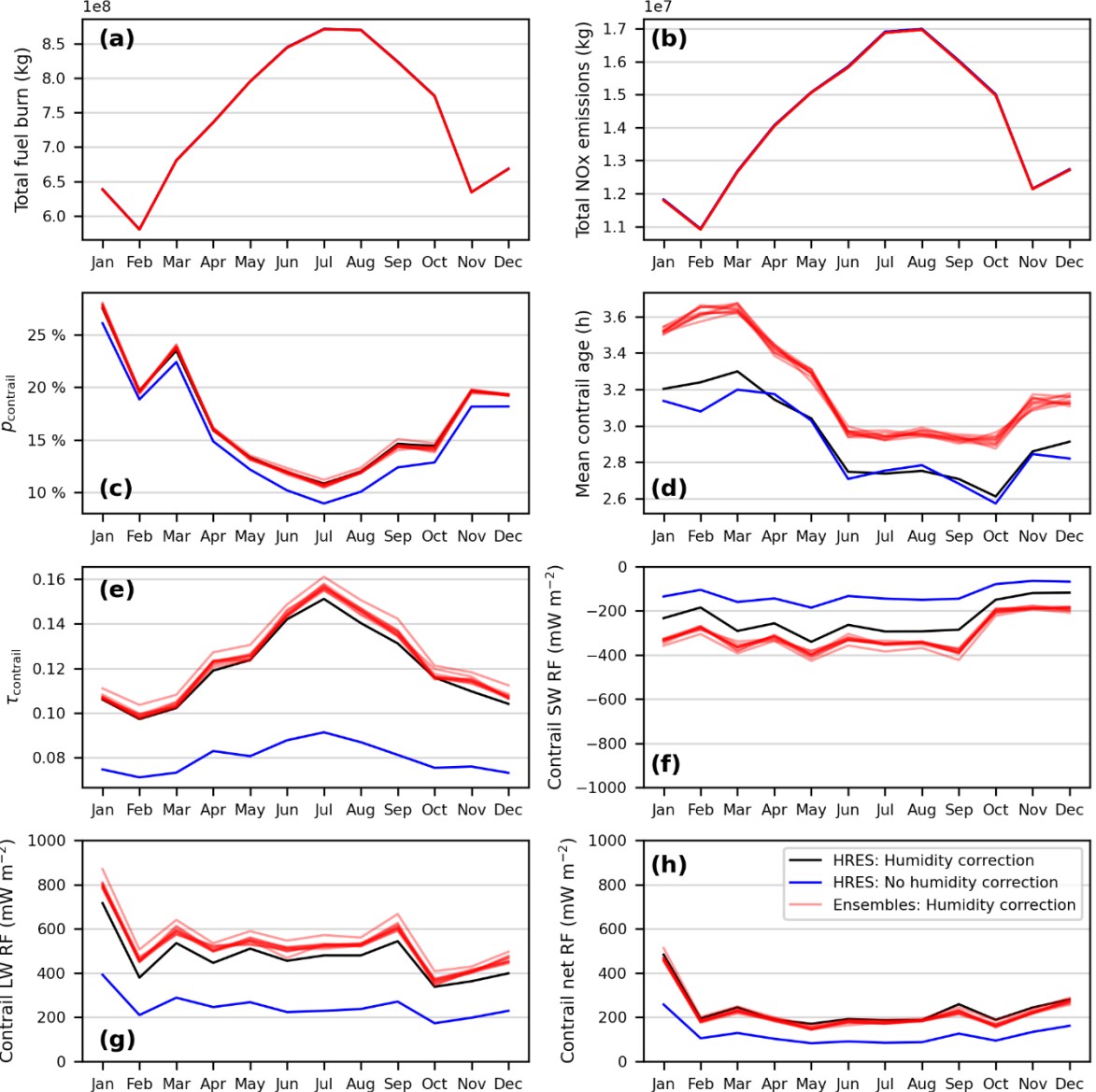

**Figure 6: Uncertainty and sensitivity analysis comparing the emissions and contrail statistics in the North Atlantic region for 2019. Contrails are simulated using meteorological data from the: ERA5 HRES with the humidity correction in Sect. 2.2 (black lines); and without humidity correction (blue lines); and the ERA5 10-member ensembles with humidity correction, where individual red lines represent the results for each ensemble member.**

### 3.5 Sensitivity analysis

To evaluate the sensitivity of contrail results to the ERA5 humidity correction, we simulated contrails for 2019 without any correction (Fig. 6 and S21). With no corrections applied, $p_{contrail}$ decreased from 16.2% in the nominal simulation to 14.7%, and there was a 34% and 8.9% reduction in $\tau_{contrail}$ (0.08 without vs. 0.12 with correction) and mean contrail age (3.2 h without





vs. 3.5 h with correction), respectively. On the other hand, corrections applied to the humidity fields increase saturation above 120% (Fig. S11), and contrails formed in these regions could have a shorter lifetime than in the simulation without humidity

correction (Fig. S24a) as the ice particles would grow to larger $r_{ice}$ and thus experience a higher sedimentation rate. When taken together, these effects caused the annual mean contrail cirrus net RF (121 mW m$^{-2}$) to be 49% smaller than in our nominal simulation (235 mW m$^{-2}$). Both results are within the range of earlier studies which estimated the North Atlantic contrail cirrus net RF to be between 100 and 360 mW m$^{-2}$ (Burkhardt and Kärcher, 2011; Chen and Gettelman, 2013; Bock and Burkhardt, 2019; Schumann and Graf, 2013), and confirms that the contrail climate forcing is highly sensitive to the humidity fields.

An assumption of a constant nvPM EI$_n$ ($10^{15}$ kg$^{-1}$) for all waypoints leads to higher fleet-aggregated contrail properties and climate forcing than in the nominal case. $p_{contrail}$ remains unchanged because the SAC does not depend on the nvPM EI$_n$ (Schumann, 1996), however the higher initial contrail ice particle number (+25% relative to the nominal simulation) led to a 3.4%, 7.4% and 14% increase in the mean contrail lifetime, $\tau_{contrail}$, and annual mean net RF (267 mW m$^{-2}$) respectively. When the comparison is made for individual flights, however, a constant nvPM EI$_n$ leads to more significant differences in the initial

ice crystal number (up to $\pm$ 3.5 $\times 10^{13}$ m$^{-1}$), $r_{ice}$ ($\pm$ 14 μm), $\tau_{contrail}$ ($\pm$ 0.78), lifetime ($\pm$ 10 h) and net RF' ($\pm$ 30 W m$^{-2}$) (Fig. S22), compared to the nominal case. These results suggest that the new ICAO EDB nvPM database, which captures the emissions profile for specific aircraft-engine types, is critical in identifying flights with the largest EF$_{contrail}$. A separate simulation assessing the aircraft mass assumptions (high/low mass at the initial waypoint), which influences the nvPM EI$_n$ and wake vortex dynamics, leads to a $\pm$ 7% sensitivity in the annual mean contrail cirrus net RF relative to the nominal simulation.

The simulated contrail outputs are less sensitive to $p_{activation}$ (c.f. Eq. (2)) and an assumption of $p_{activation} = 1$ changes the annual mean contrail cirrus net RF by +0.8% (237 mW m$^{-2}$) relative to the nominal simulation because only 24.8% of all flights form contrails near the SAC threshold temperature ($dT_{SAC} > -5$ K). For these flights, the sensitivity to $p_{activation}$ was also relatively small for other contrail properties, including initial ice crystal number (+14% on average), $r_{ice}$ (-3.7%), contrail age (+2.8%), $\tau_{contrail}$ (+5.0%) and net RF' (+0.6%) (Fig. S23).

## 4 Conclusions

We quantified aviation emissions and contrail climate forcing in the North Atlantic from January 2016 to March 2021. From 2016 to 2019, the total CO$_2$, and NO$_x$ emissions grew by 3.1% and 4.5% p.a., respectively, followed by a significant decline of 71% and 74% respectively in 2020 due to the COVID-19 pandemic. Figure 7 shows that the: (i) interannual variability in the annual mean contrail cirrus net RF (204 – 280 mW m$^{-2}$, between 2016 and 2019) is larger than the ensemble uncertainties

for 2019 (216 – 238 mW m$^{-2}$); and (ii) the contrail cirrus net RF is most sensitive to the ERA5 humidity correction, followed by the nvPM EI$_n$ and aircraft mass assumptions, and is least sensitive to $p_{activation}$. The 2016-2019 nominal contrail cirrus net RF (204 – 280 mW m$^{-2}$) from our study is larger than the reported global values (33 – 189 mW m$^{-2}$) because of the higher relative air traffic density in the North Atlantic and within the range of earlier estimates for the North Atlantic (70 – 360 mW m$^{-2}$) (Chen and Gettelman, 2013; Schumann et al., 2015; Schumann and Graf, 2013; Burkhardt and Kärcher, 2011; Bock and





Burkhardt, 2019). Our estimate is smaller than the 2006 North Atlantic estimates from Schumann & Graf (2013) (240 – 360 mW m$^{-2}$) because our study uses a larger spatial domain (Fig. S5b). However, our contrail net RF estimates increase to 281 – 386 mW m$^{-2}$ if we apply the same domain as Schumann & Graf (2013), showing consistency between the two studies.

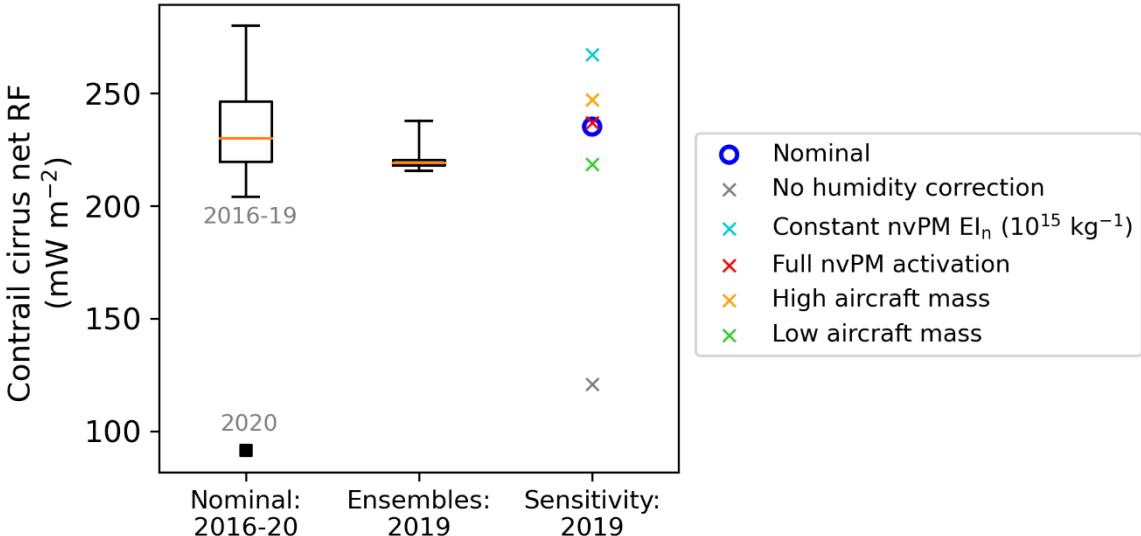

**Figure 7:** Comparison of the annual contrail cirrus net RF in the North Atlantic that is provided by: (i) the nominal simulations
using the ERA5 HRES (2016 – 2020); as well as from the (ii) ERA5 10-member ensembles; and (iii) sensitivity analyses for 2019.

The set of factors associated with strongly warming/cooling contrails can be explained by diurnal and seasonal patterns in meteorology and radiation, background cloud fields, and the nvPM emissions profile from different aircraft types, and these have implications for contrail mitigation.

On average, 12% of all flights in this region cause 80% of the annual EF$_{contrail}$ (Table 1), and this subset of flights (with large
EF$_{contrail}$ forecasts) could be supported by changes to the airline flight plan and tactical trajectory adjustments based on up-to-date meteorological forecasts. More generally, the OTS in Shanwick and Gander, which is currently designed for the safe and efficient flow of traffic based on meteorological conditions, indicative airline flight plans and other operational factors, could also be optimised pre-tactically to minimise both fuel consumption and EF$_{contrail}$ whenever possible. For example, the OTS could minimise flight distances in regions with large $dT_{SAC}$ (Fig. 4e), and above low-level water clouds with high albedo (Fig.
4j). Diversions around regions with very high ice supersaturation (RHi > 120%) might not be necessary because of a higher probability of forming contrails with shorter lifetimes (Fig. S24a). In addition to trajectory modifications, an unsophisticated approach might minimise the number of flights at selected times of the day (i.e., dusk) or season (i.e., winter) where the risk of forming strongly warming contrails is greatest (Fig. 4a and 4i).

Future research should be directed towards: (i) improvements in data assimilation and humidity representation in numerical
weather prediction models; (ii) the development of a decision-making framework that accounts for the overall climate forcing and meteorological uncertainties (Sect. 3.4 and 3.5), where flights are only diverted when their net climate benefits can be





determined with a high degree of confidence; and (iii) quantification of the effectiveness of different mitigation options proposed above.

## Author Contributions

Conceptualization, methodology and investigation, RT, US, and MEJS; Software and data curation, RT, US and MS; Visualisation, RT and MEJS; Resources, JM and GK; Writing – original draft, RT; Writing – review and editing, RT, US, EG, MS, JM, GK, CV, MEJS; Funding acquisition: MEJS; All authors have read and agreed to the published version of the manuscript.

## Competing Interests

The authors declare that they have no conflict of interest.

## Acknowledgements

EG was supported by a Royal Society University Research Fellowship (grant no. URF/R1/191602). The authors thank Susanne Rohs and the IAGOS team for providing early access to the latest IAGOS dataset, containing more flights and valid water vapour measurements. IAGOS data were created with support from the European Commission, national agencies in Germany
(BMBF), France (MESR), and the UK (NERC), and the IAGOS member institutions (http://www.iagos.org/partners). The participating airlines (Deutsche Lufthansa, Air France, Australian, China Airlines, Iberia, Cathay Pacific, Air Namibia, Sabena) supported IAGOS by carrying the measurement equipment free of charge since 1994. The data are available at http://www.iagos.fr thanks to additional support from AERIS. This document has been created with or contains elements of Base of Aircraft Data (BADA) Family 4 Release 4.2 which has been made available by EUROCONTROL to Imperial College
London. EUROCONTROL has all relevant rights to BADA. ©2019 The European Organisation for the Safety of Air Navigation (EUROCONTROL). All rights reserved. EUROCONTROL shall not be liable for any direct, indirect, incidental, or consequential damages arising out of or in connection with this document, including with respect to the use of BADA. We thank Prof Ian Poll for reviewing an earlier version of the manuscript.

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
