# Peer review of "Aviation contrail climate effects in the North Atlantic from 2016-2021"

_Atmospheric Chemistry and Physics, 2022_

## Community Comment (CC1)

Concerning the comment made by Anonymous Reviewer #2 with respect to lines 223ff. of Teoh et al.'s study, I would like to underpin that in my opinion the reviewer is fully hitting the mark here. Converting a local radiative forcing into anything like a local or regional surface temperature response by means of an equilibrium climate sensitivity parameter is at odds with the radiative forcing concept. Rather, the climate sensitivity parameter is sensibly be used to estimate contributions of global forcing components to global mean surface temperature change (see, e.g., Ramaswamy et al., 2019). The latter develops slowly in response to the radiative forcing (or a change of radiative forcing as is meant here); see, e.g., Figure 8 in Ponater et al. (2006). This "global warming" time scale is much longer than, e.g., one "COVID year".

Any local surface temperature response that might be induced, on shorter time scales, close to the location of the regional forcing cannot be derived from such global considerations. As also stated by Teoh et al., such a temperature signal is very unlikely to be observable for forcings in the order of magnitude considered here, in view of the much higher background variability of local/regional temperature. The controversial discussion of an impact of contrails on regional diurnal temperature range forms an example for the related attribution problems (Travis et al., 2002; Hong et al., 2008, Dietmüller et al., 2008; Sandhu and Baldini, 2013).

In the context of the authors' general results and discussions the surface temperature change aspect is rather circumstantial and could easily be omitted from the paper without in any way declining its merits. However, as this tendency of interpreting local radiative forcings as the direct origin of local surface temperature impact has not been uncommon in aviation climate impact studies, the authors might feel encouraged to use the opportunity for clarifying the respective issue.

References:
Dietmüller, S., et al., 2008: Contrails, natural clouds, and diurnal temperature range, J. Clim. 21, 5061-5075.
Hong, G., et al., 2008: Do contrails significantly reduce diurnal temperature range? Geophys. Res. Lett. 35, L23815.
Ponater, M., et al., 2006: Potential of the cryoplane technology to reduce aircraft climate impact: a state-of-the-art assessment, Atmos. Environ. 40, 6928-6944.
Ramaswamy, V., et al., 2019: Radiative forcing of climate: the historical evolution of the radiative forcing concept, the forcing agents and the quantification, and applications, Meteor. Monogr. 14.1-14.101.
Sandhu, A.S., Baldini, J.U.L., 2018: Evaluating the significance of the contrail effect on diurnal temperature range using the Eyjafjallajökull eruption-related flight disruption, Geophys. Res. Lett. 45, 13090-13098.
Travis, D.J., 2002: Contrails reduce diurnal temperature range, Nature 418, 601.

---

## Author Comment (AC1)

**Response to Reviewer Comments**

We thank the reviewers for their careful comments, which improved the quality of the manuscript. Below, the reviewer's comments are repeated in the *italic text*. Our response follows in normal letters. Blue text is used to cite from the revised manuscript. When page and line numbers are specified, they refer to the clean version of the revised manuscript.

**REFEREE 1 (RC1)**

*This paper aims at addressing the radiative impact of contrails in the North Atlantic between 2016 and 2021. This is an important issue since contrails represent a non-$CO_2$ effect of potentially high magnitude. In order to select appropriate mitigation strategies, comprehensive knowledge of the most significant processes is needed. To this respect this paper provides an important contribution and I strongly recommend its publication in ACP.*

*The methodology is based on the use of several blocks:*

- *Air traffic information*
- *Meteorology data over a 6 years period*
- *Aircraft type and performance, mass, engine properties, emissions*
- *A contrail-cirrus prediction tool.*

*Each set of data required the use of various tools and methodologies, for instance to estimate aircraft engine emissions for cruise conditions from ICAO LTO dataset or to apply corrections to the ERA5 humidity fields. It looks to me that this represents a tremendous amount of work and the application of a strategy/methodology following already previously published work.*

*The paper is clear and very well written. It is an impressive work.*

**Specific Comment**

1. *Line 245-250: The mean ice crystal radius is smaller in wintertime relative to the summer. Less condensable matter is available (RHi is lower in figure S14e) probably explains this point. Mentioning that contrails are formed at lower temperature ("lower temperature with less condensable water") is finally confusing since the effect of temperature is already included in the RHi which drives particle water uptake.*
   - We agree with this recommendation that the lower temperature is already accounted for in the when calculating the RHi, and have revised the sentence to improve clarity:
   [Lines 245 – 246] "a higher proportion of flights cruise above or close to the tropopause in wintertime because of the lower tropopause height (Fig. S13) and contrails are formed  with less condensable water (Fig. S14d )."

2. *I probably missed a definition of the mean radius $r_{ice}$. Is it averaged along the contrail forming flight distance?*
   - Thank you for identifying this. We have re-written the sentence to improve clarity and make clear that: (i) the acronym $r_{ice}$ is the contrail ice crystal volume mean radius; and (ii) the quantities ($r_{ice}$ and optical depth) are averaged over all contrail-forming flights in wintertime:
   [Lines 242 – 243] "The  contrail ice crystal **volume mean** radius ($r_{ice}$) and $\tau_{contrail}$ **averaged over all contrail-forming flights** in wintertime are around 25% smaller relative to the summer,  **and these are** likely caused by seasonal variations in the tropopause and temperature."

3. *Supporting information line 260: there are quite a number of papers deriving saturation water vapour pressures. Has the choice of Sonntag (1994) been made for consistency with some other data, for instance in ERA5. Surely the choice of the parameterization used can modify RHi significantly and the predicted ISSR.*

- Two derivations of saturation pressure over liquid water ($p_{liq}$) and ice ($p_{ice}$) from Sonntag (1994) and Murphy and Koop (2005) were previously compared in Appendix A3 of Schumann (2012). The attached figure (below) shows the relative difference for $p_{liq}$ and $p_{ice}$ between Sonntag (1994) and Murphy and Koop (2005).

[Figure]

- In summary:
  - the difference in $p_{ice}$ is <0.5% for temperatures > -100°C (or > 173K)
  - the difference in $p_{liq}$ is <1% for temperatures > -45°C (or >228K), and this temperature range is relevant for the Schmidt-Appleman Criterion (SAC).
  - Although differences in $p_{liq}$ can be up to 12% for temperatures between -45°C and -100°C (173K < $T$ < 228K), this temperature range is irrelevant for our applications.
- A more recent summary of the different relationships on the saturation pressure is given by Vömel (2016).
- Thank you for highlighting this. We agree with the importance of these points and have included them in the Supporting Information for the reference of future readers:
  [SI: Lines 262 – 267] "Different parameterisations of $p_{ice}$ are available (Sonntag, 1994; Murphy and Koop, 2005), and an earlier comparison between these parameterised functions showed that the differences in $p_{ice}$ is less than 0.5% for temperatures greater than -100°C (Appendix C of Schumann (2012)). A more recent assessment of the different relationships on the saturation pressure is given by Vömel (2016)."

**Technical corrections**

4. *Line 51: Missing ".". after "...Heymsfield, 2017)"*
5. *Line 78: There are actually two ICAO 2021 references in this paper and they should be called differently. One of them is misplaced in the References section (line 504).*
6. *Line 122: delete "and" after "ensembles)"*
7. *Line 425 and 427 correct Schumann & Graf*
8. *Supporting Information Line 441 Table S5 caption : spelling « annonymised »*

- Thank you for identifying these technical errors, the necessary corrections have been applied to address Points 4 to 8.

9. *Line 226: Correction "than it would have been" instead of "than it would been"*
   - Thank you. However, after considering the feedback from RC2 (Comment 11) and CC1, we have removed the results and discussion on extrapolating the change in contrail cirrus net RF to the change in regional surface temperature response.

**REFEREE 2 (RC2)**

*Teoh et al. make use of detailed flight recordings over the North Atlantic ocean in combination with a number of parameterisations to estimate various aspects of the climate effects of contrails in this region. A specific aspect of their study is the estimate of the reductions in air traffic due to the COVID-19 pandemic. The study is of interest to the readers of Atmos. Chem. Phys. It is very diligently conducted and excellently written. I have only a few minor remarks that should be addressed before publication.*

**Minor remarks**

10. *l48 The authors should discuss the IPCC AR6 assessment.*
    - The IPCC AR6 assessment on the contrail cirrus effective radiative forcing (ERF), 0.057 [0.019, 0.098] W m$^{-2}$, were cited directly from Lee et al. (2021) and draws the same conclusions.
    - Thank you for this suggestion, we have now cited the Chapter 6 of IPCC AR6 (Szopa et al., 2021) as an additional source of reference:
      [Lines 44 – 46] "To account for these second-order effects, the climate forcing of contrail cirrus is also quantified as the effective radiative forcing (ERF) (Ponater et al., 2021; Lee et al., 2021; Myhre et al., 2013; **Szopa et al., 2021**)."

11. *l223 – 225: it is unclear what are local / regional effects and what are global ones. The net cirrus RF given in l224 in my understanding is regional, for the region of interest of the North Atlantic ocean. The contrail climate sensitivity of Kärcher (2018) is a global number. To estimate the cooling effect (0.05 to 0.07 K), was some effort made to extrapolate the RF globally? Or is there a reason to believe the cooling would be confined to the region where the RF occurs? Also I have trouble seeing where the 0.05 K lower bound comes from. Isn't it rather 0.02 K?*
    - We have considered the above comment and those from CC1, and agree that our application of a global climate sensitivity parameter on a regional scale is inappropriate and lacks scientific rigour.
    - As such, any extrapolations of the change in contrail cirrus net RF to the surface temperature change have been removed from the manuscript:
      [Lines 222 – 225] "We also simulated contrails for the COVID-19 period with pre-COVID traffic to approximate the likely contrail climate forcing under normal traffic conditions and in this scenario, the annual contrail cirrus net RF increased from 69.6 mW m$^{-2}$ (actual COVID scenario) to 235 mW m$^{-2}$**, which is 15% higher relative to the pre-COVID period (204 mW m$^{-2}$)**. "

12. *l229 Where is this number seen in Table 1?*
    - The statement "around 12% of all flights in this region account for 80% of the annual EF$_{contrail}$" was derived using data from the final row of Table 1. It is the average (mean) number between 2016 and 2020, which is 11.6%, and rounded to ~12%.
    - We have revised the sentence for clarity improvements:
      [Line 226] " **On average (from 2016 to 2020),** around 12% of all flights in this region accounted for 80% of the annual EF$_{contrail}$ (Table 1 and Fig. 2)…"

13. *l262 A bit puzzling logic. The situation appears particularly frequently in summer, much more so than in winter, i.e., at times where a net cooling would be more likely.*
    - The percentage of cloud-contrail overlap, i.e., the contrail area overlapping with natural cirrus, is higher during the winter (~90%) than in the summer (~70%), meaning that contrails are more likely to be formed under cloud-free conditions during the summer with a net cooling effect.
    - Thank you for highlighting this. We acknowledge the lack of clarity and have re-written this paragraph (below). We also quantified the percentage of time periods where contrails exhibit a net cooling effect:
    [Lines 258 – 263] "An earlier study highlighted that contrails formed over the North Atlantic can exhibit a net cooling effect under cloud-free conditions (Sanz-Morère et al., 2021). However, our  **analysis of the hourly mean contrail cirrus net RF suggests that** these cooling periods  occur  infrequently **(~13% in 2019, Fig. 3) because around 70% (summer)**.  to 90% **(winter)** of the contrail area overlaps with natural cirrus  (Fig. 1d). The **mean ERA5 natural cirrus coverage in this region varies between 40% (summer) and 59% (winter), and the** contrail cirrus coverage generally peaks at around 0.7% in the summer (Fig. 1e) **because of** the minimum natural cirrus cover and cloud-contrail overlap during this period."

14. *l267 How is that possible? Shouldn't there be more incoming sunlight in summer (cf Fig. S14i)?*
    - Initially, we were also expecting the SDR to be higher in the summer than in the spring because of the longer daylight hours. However, the mean SDR (derived from the ERA5 HRES) in spring and summer over the North Atlantic are both estimated to be ~394 W m$^{-2}$ and the longer daylight hours during the summer has been accounted for.
    - We attribute this to the specific dates/cut-off points that were used to define the beginning of each season:
        - Spring: 20-March (~393.6 W m$^{-2}$)
        - Summer: 21-June (~393.7 W m$^{-2}$)
        - Autumn: 22-Sept (~127.8 W m$^{-2}$)
        - Winter: 21-Dec (~126.5 W m$^{-2}$)

    Figure S14i is replotted (below) and included in the Supporting Information. The shaded regions with different colours, which shows the cut-off points for each of the four seasons, will help to explain the phenomenon of both seasons (spring and summer) having a comparable mean SDR of ~394 W m$^{-2}$.

[Figure]

15. *l282 EF was introduced as an integral measure, why would one now normalize again by contrail length? Why not length and width and go for forcing?*
    - The different normalisations of EF$_{contrail}$ are used to provide different insights depending on the research question. In this manuscript, we used the:
        - (i) **EF$_{contrail}$**, which quantifies the total contrail climate forcing throughout the lifetime of each contrail segment, flight (Figure 4), or time period (Figure 2),

(ii) **EFcontrail per flight distance**, which was used in Figure 5 to evaluate the differences in contrail climate forcing that arise from different aircraft types (with different nvPM $EI_n$). For example, under the same meteorological conditions, we would expect the $EF_{contrail}$ per flight distance to be larger for aircraft types with a higher nvPM $EI_n$, and

(iii) **EFcontrail per contrail length**, which was used in Figure 3b to account for differences in air traffic density and meteorological conditions; and to isolate the diurnal/seasonal effects. For example, Figure 3b shows that contrails forming at night, per unit length, are expected to have an $EF_{contrail}$ that is up to one order of magnitude larger than those that were formed during the day. If we were to use the $EF_{contrail}$ metric, the results could show a smaller total $EF_{contrail}$ value during the night because of the lower air traffic activity where less contrails are formed.

16. *l285 Fig. 3. The legend says "time of day" but really it is UTC, isn't it? There is a large ambiguity on which time (in UTC) is sunrise and which time is sunset, given the breadth in longitude of the Atlantic Ocean. Else it would be useful to indicate the time (spans) of sunrise and sunset that are discussed in the text. As written above, I do not understand the usefulness of EF per length, why not stick to EF, or else omit this panel.*

- Thank you. We agree with this comment and have changed the x-label of Figure 3 from "Time of day (UTC)" to "Coordinated Universal Time, UTC". We also applied this modification to all other figures in the manuscript and Supporting Information with the term "Time of day".

- To identify the specific sunrise and sunset times in this region, we have also included Fig. S16 in the Supporting Information (also attached below) that shows the hourly mean solar direct radiation (SDR) over the North Atlantic for each day in 2019 to complement Fig. 3 in the main text (also shown below).

[Figure]

- The rationale for selecting the $EF_{contrail}$ per contrail length over the absolute $EF_{contrail}$ in Fig. 3b is addressed in Comment (15).

17. *l321 same comment as above on SDR*

- Thank you. The comparable mean SDR in both spring and summer (~394 W m$^{-2}$) is valid and addressed in Comment (14).

- We have also cross-referenced Fig. S16 in Line 320 (manuscript), also shown in Comment (16), to highlight the short daylight hours and minimum SDR in wintertime over the North Atlantic.

18. *l336 The half-sentence "below optically thick high-level cirrus" should better start a new sentence with the second argument/condition. But this is not so obvious. If the cirrus are optically thick, why would they not have the same effect in the solar as the optically thick low-level clouds?*

- We note that TOA irradiances are different over cloudy and clear land surfaces and different over low-level clouds and mid-tropospheric clouds or high clouds. Outgoing longwave radiation OLR is smaller over cloudy domains because of lower brightness temperature. The difference is small over low-level clouds. The reflected solar radiation (RSR) is higher over cloudy areas than over clear surfaces because the clouds enhance solar radiation scattering. The contrail RF values respond to the different OLR and RSR values, to the temperature and emissivity, and to the solar backscatter by the contrail cloud. Therefore, the climate forcing effects between low- and high-level cirrus and over clear and clouded surface are not the same. The different effects are covered by the CoCiP radiative forcing model which computes the local RF values as a function of the ERA5-given TOA irradiances, contrail properties and cirrus optical depth above the contrails ($\tau_{cirrus}$) (Schumann et al., 2012).

- We thank the reviewer for these suggestions and have re-written the paragraph to improve clarity (below):
  [Lines 333 – 342] "In contrast, strongly cooling contrails are more common over regions with little low-level cloud,  **where** a strong albedo contrast with the dark ocean surface (mean underlying albedo of 0.29, Fig. 4j) **leads to a maximum SW RF'**.  **S**trongly cooling contrails **are also more likely when formed below high-level cirrus with a higher** mean overlying natural cirrus optical depth ($\tau_{cirrus}$) **of 0.17, which** is around two times larger than for strongly warming contrails ( 0.081) (Fig. 4k). **This is because for high-level cirrus, albedo (driving SW RF) depends less strongly on optical depth compared to the dependence of emissivity (driving LW RF) on the optical depth (Wallace and Hobbs, 2006). Therefore, a higher $\tau_{cirrus}$ increases the emissivity of the overlying high-level cirrus and reduces the LW RF' that can be attributed to the underlying contrail, while the smaller rate of increase in cirrus albedo (relative to its emissivity) allows some incoming solar radiation to reach the underlying contrail, and that the contrail SW RF' would be reduced by a smaller degree relative to the reduction in its LW RF'** ."

19. *l347 It is noteworthy perhaps that the cirrus that are neither strongly cooling nor strongly warming seemingly have a smaller absolute effect in either direction, they occur at smaller nvPM.*

- Thank you for this suggestion. We have added a sentence in the manuscript to highlight this point:
  [Lines 362 – 363] "In contrast, contrails formed from aircraft types with smaller nvPM emissions are neither strongly cooling nor strongly warming (Fig. 5d)."

20. *l371 Is there an explanation for this result? Is there a reason to believe the HRES resolution is better? Is it appropriate, or would still higher resolution lead to still smaller results?*

- The ERA5 HRES has a spatiotemporal resolution of $0.25° \times 0.25° \times 37$ pressure levels $\times$ 1 h and is the best available dataset that we currently have access to. In contrast, the ERA5 10-member ensemble has a resolution of $0.5° \times 0.5° \times 37$ pressure levels $\times$ 3 h.

- We would expect a higher resolution meteorological dataset to be able to capture more accurately the: (i) fine-scale structure of ice-supersaturated regions; and (ii) sub-grid scale variability in the RHi. Therefore, we would expect the simulated contrail properties and climate forcing to change if the spatiotemporal resolution of the

meteorological dataset is further increased. However, any guesses on the spatiotemporal resolution leading to smaller/larger results would be speculative at this stage.

21. *l381 Is there some problem in the parameterisation that leads to this strong increase in contrail age simply because the input fields have a coarser temporal resolution?*

- This is an interesting question. The ERA5 10-member ensemble is made up of: (i) one control member that is also used to initialise the ERA5 HRES; and (ii) nine perturbed members where are initialised with random perturbations added to the observations (Hersbach et al., 2020) [Lines 208 – 211 in the SI].
- If we compare at the simulated mean contrail age from the ERA5 10-member ensemble (Table S6 in the SI) relative to the nominal simulation using the ERA5 HRES (Table 1 in the main text), all 10 ensemble members, including the control member that is also used to initialise the ERA5 HRES, consistently show a larger contrail age relative to the nominal simulation.
- This result suggests that differences in spatiotemporal resolution between the ERA5 10-member ensemble ($0.5° \times 0.5° \times 37$ levels $\times 3$ h) and ERA5 HRES ($0.25° \times 0.25° \times 37$ levels $\times 1$ h) is likely the main contributing factor to the larger simulated contrail age, than potential differences in parameterisations/perturbations. A lower spatiotemporal resolution of the ERA5 10-member ensemble implies that it will be less capable of capturing sub-grid areas where RHi < 1 relative to the HRES. Therefore, contrails simulated using the HRES are likely to have shorter lifetimes relative to the ensembles because it is more likely to encounter pockets of air mass where atmospheric conditions are subsaturated.
- Thank you for highlighting this, we have updated the manuscript to include this explanation:
  [Lines 381 – 386] "The characteristics of flights with strongly warming/cooling contrails in each ensemble member (Fig. S21) are generally consistent with the HRES (Fig. 4). However, unlike the HRES, the ensembles do not predict the occurrence of strongly warming or cooling contrails before dawn or dusk, respectively (Fig. 4i vs. Fig. S21i). This is likely due to the lower **spatio**temporal resolution of the ensembles ( relative to the HRES, **making it less capable of capturing sub-grid areas where RHi < 1**, **thereby**  causing an overprediction of the mean contrail age (+7.7% relative to the HRES, Fig. 6d) and change in the sign of EF$_{contrail}$ as contrails persist through dawn/dusk."

22. *l414 At the end of this uncertainty / sensitivity section, it would have been nice to do an overall uncertainty quantification by propagating all uncertainties to an overall uncertainty on the assessed RF.*

*l443 This could be a point where the overall uncertainty is reported.*

- Thank you for this suggestion. To propagate all the uncertainties/sensitivities to quantify an overall uncertainty on the simulated contrail cirrus net RF would require a Monte Carlo simulation with larger number of simulations to account for the different permutations, and we currently have constraints in computational resources. As a reference point, it took ~2 days to complete the contrail simulation for one-year and ~3 weeks to complete the contrail simulation with the ERA5 10-member ensemble.
- Work is currently ongoing to perform our contrail simulations using cloud computing resources, and we have added a sentence in the manuscript to identify this suggestion as a topic for future research:
- [Lines 452 – 454] "Future research should be directed towards: (i) quantifying an overall uncertainty on the simulated contrail climate forcing by propagating all the

uncertainties/sensitivities from different input parameters, including meteorology, nvPM emissions and aircraft mass assumptions."

- We have also re-written the paragraph describing Figure 7 to include a more comprehensive discussion on the ERA5 humidity correction factor, which contributes to the largest uncertainty/sensitivity to our relative to other input parameters:
  [Lines 423 – 436] "Figure 7 shows that the: (i) interannual variability in the annual mean contrail cirrus net RF (204 – 280 mW m$^{-2}$, between 2016 and 2019) is larger than the ensemble uncertainties for 2019 (216 – 238 mW m$^{-2}$); and (ii) the contrail cirrus net RF is most sensitive to the ERA5 humidity correction, followed by the nvPM EIn and aircraft mass assumptions, and is least sensitive to $p_{activation}$. The 2016-2019 nominal contrail cirrus net RF (204 – 280 mW m$^{-2}$) from our study is larger than the  **range of global values reported in previous studies** (33 – 189 mW m$^{-2}$) because of the higher relative air traffic density in the North Atlantic and within the range of earlier estimates for the North Atlantic (70 – 360 mW m$^{-2}$) (Chen and Gettelman, 2013; Schumann et al., 2015; Schumann and Graf, 2013; Burkhardt and Kärcher, 2011; Bock and Burkhardt, 2019). Our estimate is smaller than the 2006 North Atlantic estimates from Schumann and Graf (2013) (240 – 360 mW m$^{-2}$) because our study uses a larger spatial domain (Fig. S5b). However, our contrail net RF estimates increase to 281 – 386 mW m$^{-2}$ if we apply the same domain as Schumann and Graf (2013), showing consistency between the two studies. **Figure 7 also shows that the contrail cirrus net RF is most sensitive to the ERA5 humidity correction, followed by the nvPM EI$_n$ and aircraft mass assumptions, and is least sensitive to $p_{activation}$. Without correction of the humidity fields, the estimated contrail cirrus net RF is halved relative to the simulation where correction to humidity is applied. However, our analysis of in-situ humidity measurements and the known limitations of the ERA5 products (Sect. 2.2 and SI §S3) gives confidence that the uncertainty in contrail cirrus net RF is more accurately characterised by the simulations only when humidity correction is applied.**"

Typos

23. l191 "ensemble"
24. l328 "persist"
   - Thank you for identifying these typing errors, the necessary corrections have been applied to address Points 23 and 24.

25. l226 "would have been"
   - After considering the feedback from Comment 11 and CC1, we have removed the results and discussion on extrapolating the change in contrail cirrus net RF to the change in regional surface temperature response.

**COMMUNITY 1 (CC1)**

*Concerning the comment made by Anonymous Reviewer #2 with respect to lines 223ff. of Teoh et al.'s study, I would like to underpin that in my opinion the reviewer is fully hitting the mark here. Converting a local radiative forcing into anything like a local or regional surface temperature response by means of an equilibrium climate sensitivity parameter is at odds with the radiative forcing concept. Rather, the climate sensitivity parameter is sensibly be used to estimate contributions of global forcing components to global mean surface temperature change (see, e.g., Ramaswamy et al., 2019). The latter develops slowly in response to the radiative forcing (or a change of radiative forcing as is meant here); see, e.g., Figure 8 in Ponater et al. (2006). This "global warming" time scale is much longer than, e.g., one "COVID year".*

*Any local surface temperature response that might be induced, on shorter time scales, close to the location of the regional forcing cannot be derived from such global considerations. As also stated by Teoh et al., such a temperature signal is very unlikely to be observable for forcings in the order of magnitude considered here, in view of the much higher background variability of local/regional temperature. The controversial discussion of an impact of contrails on regional diurnal temperature range forms an example for the related attribution problems (Travis et al., 2002; Hong et al., 2008, Dietmüller et al., 2008; Sandhu and Baldini, 2013).*

*In the context of the authors' general results and discussions the surface temperature change aspect is rather circumstantial and could easily be omitted from the paper without in any way declining its merits. However, as this tendency of interpreting local radiative forcings as the direct origin of local surface temperature impact has not been uncommon in aviation climate impact studies, the authors might feel encouraged to use the opportunity for clarifying the respective issue.*

*References:*

*Dietmüller, S., et al., 2008: Contrails, natural clouds, and diurnal temperature range, J. Clim. 21, 5061-5075.*

*Hong, G., et al., 2008: Do contrails significantly reduce diurnal temperature range? Geophys. Res. Lett. 35, L23815.*

*Ponater, M., et al., 2006: Potential of the cryoplane technology to reduce aircraft climate impact: a state-of-the-art assessment, Atmos. Environ. 40, 6928-6944.*

*Ramaswamy, V., et al., 2019: Radiative forcing of climate: the historical evolution of the radiative forcing concept, the forcing agents and the quantification, and applications, Meteor. Monogr. 14.1-14.101.*

*Sandhu, A.S., Baldini, J.U.L., 2018: Evaluating the significance of the contrail effect on diurnal temperature range using the Eyjafjallajökull eruption-related flight disruption, Geophys. Res. Lett. 45, 13090-13098.*

*Travis, D.J., 2002: Contrails reduce diurnal temperature range, Nature 418, 601.*

- We thank CC1 for the detailed explanation and agree that our application of a global climate sensitivity parameter on a regional scale were overly simplistic and speculative. As such, we have removed any extrapolations and discussions on the surface temperature change from the manuscript:
- [Lines 222 – 225] "We also simulated contrails for the COVID-19 period with pre-COVID traffic to approximate the likely contrail climate forcing under normal traffic conditions and in this scenario, the annual contrail cirrus net RF increased from 69.6 mW m$^{-2}$ (actual COVID scenario) to 235 mW m$^{-2}$, **which is 15% higher relative to the pre-COVID period (204 mW m$^{-2}$)**. "

**References**

Bock, L. and Burkhardt, U.: Contrail cirrus radiative forcing for future air traffic, Atmos. Chem. Phys., 19, 8163–8174, https://doi.org/10.5194/acp-19-8163-2019, 2019.

Burkhardt, U. and Kärcher, B.: Global radiative forcing from contrail cirrus, Nat. Clim. Chang., 1, 54–58, https://doi.org/10.1038/nclimate1068, 2011.

Chen, C. C. and Gettelman, A.: Simulated radiative forcing from contrails and contrail cirrus, Atmos. Chem. Phys., 13, 12525–12536, https://doi.org/10.5194/acp-13-12525-2013, 2013.

Hersbach, H., Bell, B., Berrisford, P., Hirahara, S., Horányi, A., Muñoz-Sabater, J., Nicolas, J., Peubey, C., Radu, R., Schepers, D., Simmons, A., Soci, C., Abdalla, S., Abellan, X., Balsamo, G., Bechtold, P., Biavati, G., Bidlot, J., Bonavita, M., De Chiara, G., Dahlgren, P., Dee, D., Diamantakis, M., Dragani, R., Flemming, J., Forbes, R., Fuentes, M., Geer, A., Haimberger, L., Healy, S., Hogan, R. J., Hólm, E., Janisková, M., Keeley, S., Laloyaux, P., Lopez, P., Lupu, C., Radnoti, G., de Rosnay, P., Rozum, I., Vamborg, F., Villaume, S., and Thépaut, J. N.: The ERA5 global reanalysis, Q. J. R. Meteorol. Soc., https://doi.org/10.1002/qj.3803, 2020.

Kärcher, B.: Formation and radiative forcing of contrail cirrus, Nat. Commun., 9, 1824, https://doi.org/10.1038/s41467-018-04068-0, 2018.

Lee, D. S., Fahey, D. W., Skowron, A., Allen, M. R., Burkhardt, U., Chen, Q., Doherty, S. J., Freeman, S., Forster, P. M., Fuglestvedt, J., Gettelman, A., De León, R. R., Lim, L. L., Lund, M. T., Millar, R. J., Owen, B., Penner, J. E., Pitari, G., Prather, M. J., Sausen, R., and Wilcox, L. J.: The contribution of global aviation to anthropogenic climate forcing for 2000 to 2018, Atmos. Environ., 244, 117834, https://doi.org/10.1016/J.ATMOSENV.2020.117834, 2021.

Murphy, D. M. and Koop, T.: Review of the vapour pressures of ice and supercooled water for atmospheric applications, Q. J. R. Meteorol. Soc., 131, 1539–1565, https://doi.org/10.1256/QJ.04.94, 2005.

Myhre, G., Shindell, D., Bréon, F.-M., Collins, W., Fuglestvedt, J., Huang, J., Koch, D., Lamarque, J.-F., Lee, D., Mendoza, B., Nakajima, T., Robock, A., Stephens, G., Takemura, T., and Zhang, H.: Anthropogenic and Natural Radiative Forcing. In: Climate Change 2013: The Physical Science Basis. Contribution of Working Group I to the Fifth Assessment Report of the Intergovernmental Panel on Climate Change, Cambridge University Press, Cambridge, United Kingdom, 665 pp., 2013.

Ponater, M., Bickel, M., Bock, L., and Burkhardt, U.: Towards Determining the Contrail Cirrus Efficacy, 8, 42, https://doi.org/10.3390/AEROSPACE8020042, 2021.

Sanz-Morère, I., Eastham, S. D., Allroggen, F., Speth, R. L., and Barrett, S. R. H.: Impacts of multi-layer overlap on contrail radiative forcing, Atmos. Chem. Phys., 21, 1649–1681, https://doi.org/10.5194/ACP-21-1649-2021, 2021.

Schumann, U.: A contrail cirrus prediction model, Geosci. Model Dev., 5, 543–580, https://doi.org/10.5194/gmd-5-543-2012, 2012.

Schumann, U. and Graf, K.: Aviation-induced cirrus and radiation changes at diurnal timescales, J. Geophys. Res. Atmos., 118, 2404–2421, https://doi.org/10.1002/jgrd.50184, 2013.

Schumann, U., Mayer, B., Graf, K., and Mannstein, H.: A parametric radiative forcing model for contrail cirrus, J. Appl. Meteorol. Climatol., 51, 1391–1406, https://doi.org/10.1175/JAMC-D-11-0242.1, 2012.

Schumann, U., Penner, J. E., Chen, Y., Zhou, C., and Graf, K.: Dehydration effects from contrails in a coupled contrail–climate model, Atmos. Chem. Phys., 15, 11179–11199, https://doi.org/10.5194/acp-15-11179-2015, 2015.

Sonntag, D.: Advancements in the field of hygrometry, Meteorol. Zeitschrift, 3, 51–66, https://doi.org/10.1127/metz/3/1994/51, 1994.

Szopa, S., Naik, V., Adihikary, B., Artaxo, P., Berntsen, T., Collins, W., Fuzzi, S., Gallardo, L., Kiendler-Scharr, A., Klimont, Z., Liao, H., Unger, N., and Zanis, P.: Short-Lived Climate Forcers. In Climate Change 2021: The Physical Science Basis. Contribution of Working Group I to the Sixth Assessment Report of the Intergovernmental Panel on Climate Change, Cambridge, United Kingdom and New York, NY, USA, 817–922 pp., https://doi.org/10.1017/9781009157896.008, 2021.

Vömel, H.: Saturation vapor pressure formulations, National Center for Atmospheric Research, Earth Observing Laboratory. Boulder, CO, USA, 2016.

Wallace, J. M. and Hobbs, P. V.: Atmospheric Science: An Introductory Survey, Second Edition., Elsevier Academic Press, Amsterdam, 1–488 pp., https://doi.org/10.1016/C2009-0-00034-8, 2006.